# Integrin adhesome axis inhibits the RPM-1 ubiquitin ligase signaling hub to regulate growth cone and axon development

Jonathan Amezquita[1,2☉], Muriel Desbois[3☉], Karla J. Opperman[1], Joseph S. Pak[1], Elyse L. Christensen[2], Nikki T. Nguyen[1], Karen Diaz-Garcia[1], Melissa A. Borgen[4], Brock Grill[1,5,6]*

1 Center for Integrative Brain Research, Seattle Children's Research Institute, Seattle, Washington, United States of America, 2 Molecular and Cellular Biology Graduate Program, University of Washington, Seattle, Washington, United States of America, 3 School of Life Sciences, Keele University, Keele, Staffordshire, United Kingdom, 4 Florida Institute of Technology, Department of Biomedical Engineering and Sciences, Melbourne, United States of America, 5 Department of Pediatrics, University of Washington School of Medicine, Seattle, Washington, United States of America, 6 Department of Pharmacology, University of Washington School of Medicine, Seattle, Washington, United States of America

☉ These authors contributed equally to this work.
* brock.grill@seattlechildrens.org

**Data Availability Statement:** All data are in the manuscript, Supporting information files, and the PRIDE database (see Methods for further details).

## Abstract

Integrin signaling plays important roles in development and disease. An adhesion signaling network called the integrin adhesome has been principally defined using bioinformatics and cell-based proteomics. To date, the adhesome has not been studied using integrated proteomic and genetic approaches. Here, proteomic studies in *C. elegans* identified physical associations between the RPM-1 ubiquitin ligase signaling hub and numerous adhesome components including Talin (TLN-1), Kindlin (UNC-112) and β-integrin (PAT-3). *C. elegans* RPM-1 is orthologous to human MYCBP2, a prominent player in nervous system development recently associated with a neurodevelopmental disorder. After curating and updating the conserved *C. elegans* adhesome, we identified an adhesome subnetwork physically associated with RPM-1 that has extensive links to human neurobehavioral abnormalities. Using neuron-specific, CRISPR loss-of-function strategies, we demonstrate that a PAT-3/UNC-112/TLN-1 adhesome axis regulates axon termination in mechanosensory neurons by inhibiting RPM-1. Developmental time-course studies and pharmacological results suggest TLN-1 inhibition of RPM-1 affects growth cone collapse and microtubule dynamics during axon outgrowth. These results indicate the PAT-3/UNC-112/TLN-1 adhesome axis restricts RPM-1 signaling to ensure axon outgrowth is terminated in a spatially and temporally accurate manner. Thus, our findings orthogonally validate the adhesome using an organismal setting, identify an adhesome axis that inhibits RPM-1 (MYCBP2), and highlight important new links between the adhesome and brain disorders.

**Funding:** B.G. was supported by National Institutes of Health Grant R01 NS072129 from the National Institute of Neurological Disorders and Stroke (NINDS). B.G. and J.A. wereas supported by NINDS Diversity Supplement: R01 NS072129-S1. B.G and J.S.P. wereas supported by the National Institute on Drug Abuse Diversity Supplement: R01 DA048036-S1. The funders had no role in study design, data collection and analysis, decision to publish, or preparation of the manuscript.

**Competing interests:** The authors have declared that no competing interests exist.

## Author summary

The adhesome is an important signaling network in development that has been principally defined using bioinformatics and proteomics with cell-lines. We have used *C. elegans* as a whole animal model to orthogonally identify and evaluate an adhesome subnetwork that regulates axon development. Our studies demonstrate that an Integrin/Kindlin/Talin adhesome axis influences axon development by regulating termination of axon outgrowth. Mechanistically, this occurs via the Integrin/Kindlin/Talin axis inhibiting the RPM-1 ubiquitin ligase signaling hub to influence microtubule and growth cone dynamics. RPM-1 and its human homolog MYCBP2 are prominent evolutionarily conserved regulators of nervous system development and associated with a neurodevelopmental disorder. Our findings validate the adhesome using an organismal setting, identify an adhesome axis that inhibits RPM-1 (MYCBP2), and highlight extensive genetic links between the adhesome and abnormal human neurodevelopment.

## Introduction

In the nervous system, integrin receptors and their signaling mechanisms have broad functions in cell migration, axon development, synaptic connectivity and axon regeneration [1–4]. Integrin signaling has been linked to neurological conditions such as autism spectrum disorder (ASD) and Alzheimer's disease [2,3]. Outside the nervous system, integrin signaling plays prominent roles in tissue development, the immune system, and cancer [5–7].

Integrins are transmembrane receptors in a large signaling and adhesion network referred to as the integrin adhesome [8–11]. The integrin adhesome facilitates cellular interactions with the extracellular matrix (ECM) by linking ECM components to intracellular F-actin thereby influencing cell migration and cellular process outgrowth. In the nervous system, integrin signaling figures prominently in neurite and axon outgrowth and guidance [12–16]. Talin and Kindlin are two core components of the adhesome which bind β-integrin receptors [17,18]. Prior studies using cultured neurons have shown Kindlin and upstream regulators of Talin, such as Calpain, regulate axon outgrowth [19,20]. At present, studies that genetically impair Talin or Kindlin directly and examine effects on axon and growth cone development *in vivo* remain absent. Moreover, the identification of the adhesome is primarily based on bioinformatics and proteomics [8]. We are unaware of studies that have integrated proteomics and genetics to evaluate how multiple adhesome components influence neuronal development in an organismal setting. An invertebrate genetic model organism, such as the nematode *C. elegans*, is ideal for such studies because of its genetic tractability and simplified integrin signaling with a single gene encoding β-integrin (PAT-3), Talin (TLN-1) and Kindlin (UNC-112).

While it is essential for axons to grow and be successfully guided to target cells, it is also critical that outgrowth be terminated—a process called axon termination [21,22]. Efficient, precise nervous system construction requires spatially and temporally accurate axon termination. For example, proper axon termination is required to generate axonal tiling patterns that occur in laminated termination zones in the spinal cord, cortical layers and retina. The importance of understanding axon termination is further highlighted by ongoing efforts to comprehensively map axon termination patterns in the rodent brain [23]. At present, it remains unclear whether adhesome signaling influences axon termination.

One of the most prominent players in axon termination identified using *C. elegans* is RPM-1 [24–27]. *C. elegans* RPM-1 is a ubiquitin ligase signaling hub that is orthologous to human

MYCBP2 [24,28–30]. MYCBP2 (also called Phr1 in rodents) is an evolutionarily conserved regulator of axon development [31,32], and genetic variants in *MYCBP2* were recently shown to cause a neurodevelopmental spectrum disorder called *MYCBP2*-related developmental delay with corpus callosum defects (MDCD) [33]. Despite the importance of RPM-1/MYCBP2 in nervous system development and disease, we know very little about how RPM-1 is regulated during growth cone and axon development. Moreover, physical and genetic interactions between RPM-1 and the adhesome have not been examined in any system to our knowledge.

Using unbiased, affinity purification (AP) proteomics, we identified numerous conserved components of the *C. elegans* integrin adhesome physically associated with RPM-1. Using an updated version of the conserved *C. elegans* adhesome, we identified genetic links to human neurobehavioral abnormalities for 75% of the components in the adhesome subnetwork associated with RPM-1. CRISPR-based, cell-specific loss of function strategies demonstrated that impairing the PAT-3/UNC-112/TLN-1 adhesome axis results in premature axon termination in mechanosensory neurons. Genetic interaction studies, CRISPR-based imaging, and developmental time-course results suggest that this β-integrin/Talin/Kindlin axis inhibits RPM-1 thereby affecting microtubule stability to influence growth cone collapse and axon termination *in vivo*.

## Results

### AP-proteomics identifies physical associations between numerous integrin adhesome components and RPM-1

To identify proteins physically associated with the RPM-1 ubiquitin ligase signaling hub, we previously performed large-scale, unbiased AP-proteomics using *C. elegans* [26, 34]. We relied upon integrated transgenes that express RPM-1 fused with a Protein G::Streptavidin binding peptide (GS) affinity tag expressed using the native *rpm-1* promoter on an *rpm-1* protein null background. Negative control animals expressed an integrated transgene where the GS tag was fused to GFP (GS::GFP) and expressed by the *rpm-1* promoter. AP-proteomics was performed with two RPM-1 constructs: GS::RPM-1 and GS::RPM-1 ligase-dead (LD). RPM-1 LD is point mutated in its RING ubiquitin ligase domain, which inactivates catalytic ligase activity and enriches ubiquitination substrates [26,35]. We previously established two groups of RPM-1 associated proteins identified by AP-proteomics, RPM-1 binding proteins and RPM-1 ubiquitination substrates (Fig 1A) [26,35]. RPM-1 binding proteins (*e.g.* RAE-1, GLO-4, FSN-1 and PPM-2) are present in both the GS::RPM-1 and GS::RPM-1 LD samples compared to the GS::GFP negative control (Fig 1B) [26,35–38]. In contrast, RPM-1 LD biochemically 'traps' and enriches ubiquitination substrates (*e.g.* ULK/UNC-51 and CDK-5) (Fig 1C) [26,35].

Here, we further analyzed our large-scale proteomics dataset, and identified numerous components of the integrin adhesome physically associated with RPM-1 (Fig 1B–1F). While a prior bioinformatic study demonstrated that the adhesome is conserved in *C. elegans* [39], we curated and updated the *C. elegans* adhesome based on several criteria: 30% or greater protein sequence similarity to mammalian adhesome components, 20% or greater sequence identity, and designation as top BLAST hit on the Alliance of Genome Resources database [40]. Our analysis identified 132 mammalian adhesome components in *C. elegans* with 123 components having unique nematode orthologs (S1 Table). Furthermore, gene ontology (GO) analysis of RPM-1 proteomic hits demonstrated that multiple adhesome components are present in the top 20 GO processes identified (S1 Fig).

In total, we identified 32 *C. elegans* adhesome components enriched more than 1.5 times in GS::RPM-1 samples compared to GS::GFP negative control samples (Fig 1B, 1D–1F; Table 1; S1 Data). Adhesome components were not enriched in the GS::RPM-1 LD sample

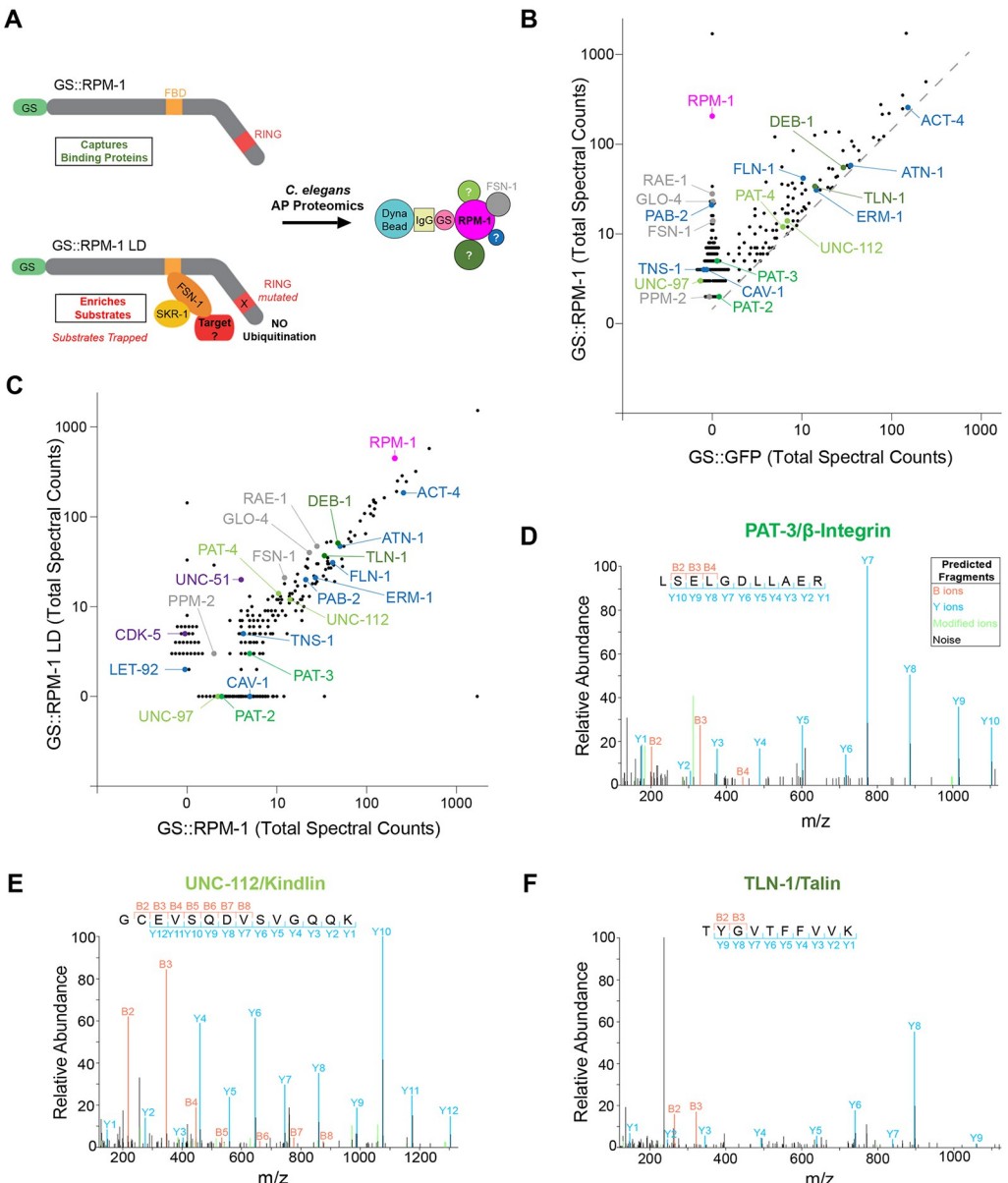

**Fig 1. RPM-1 AP-proteomics identifies numerous integrin adhesome components. A)** Schematic of RPM-1 ubiquitin ligase constructs used for AP-proteomics from *C. elegans*. Both GS::RPM-1 and GS::RPM-1 LD capture binding proteins (e.g. FSN-1), while GS::RPM-1 LD enriches ubiquitination substrates. **B)** Example of single AP-proteomics experiment showing individual proteins identified by LC-MS/MS. Shown are results from GS::RPM-1 sample compared with GS::GFP (negative control). Highlighted (shades of green) are integrin receptors, PAT-3/β-Integrin and PAT-2/α-Integrin, and two canonical integrin signaling complexes, UNC-112/Kindlin complex and TLN-1/Talin complex. Additional adhesome components (blue) were identified. Also highlighted are RPM-1 (magenta) and known RPM-1 binding proteins (gray). Grey dashed line represents 1.5x enrichment of spectra for proteomic hits in GS::RPM-1 sample compared to GS::GFP negative control. **C)** Results from single AP-proteomics experiment showing GS::RPM-1 compared to GS::RPM-1 LD ubiquitination substrate 'trap'. Highlighted are previously validated substrates (CDK-5 and UNC-51, purple). **D-F)** Examples of LC-MS/MS peptide spectrum for **D)** PAT-3/β-Integrin, **E)** UNC-112/Kindlin and **F)** TLN-1/Talin.

compared to the GS::RPM-1 sample (Fig 1C, Table 1). These results suggest that the majority of adhesome components associated with RPM-1 are not likely to be ubiquitination substrates. This included 1) the β-integrin PAT-3 and the α-integrin PAT-2; 2) the Kindlin complex composed of UNC-112/Kindlin, PAT-4/ILK, PAT-6/PARVA and UNC-97/LIMS, and 3) the Talin complex that contains TLN-1/Talin and DEB-1/Vinculin (Fig 1B–1F; Table 1; Fig 2A). Several further adhesome components were also identified (Fig 1B and 1C; Table 1). Analysis of results from 7 independent proteomic experiments showed that the majority of adhesome components were identified in two or more experiments with several significantly enriched in GS::RPM-1 samples compared to GS::GFP negative controls (Table 1; S2 Data). Interestingly, significantly enriched adhesome components included the sole *C. elegans* β-integrin PAT-3 (13% sequence coverage, S2 Fig), the sole Kindlin UNC-112 (35% sequence coverage, S3 Fig), and the sole Talin TLN-1 (28% sequence coverage, S4 Fig). Thus, in vivo

**Table 1. Summary of adhesome components detected in RPM-1 AP-proteomics using *C. elegans*.** Shown are cumulative results from seven independent RPM-1 AP-proteomics experiments. Reported are total peptide spectra for adhesome components enriched 1.5x in either GS::RPM-1 or GS::RPM-1 LD over GS::GFP negative control samples. Note adhesome components do not show more than 2 fold increase in GS::RPM-1 LD substrate 'trap' compared to GS::RPM-1 samples suggesting they are not likely to be ubiquitination substrates. Significance determined using Mann-Whitney test (*p*-value) and corrected for multiple comparisons using *post hoc* Benjamini-Hochberg method (*q*-value) with a 5% false discovery rate (FDR). For statistical comparison and fold enrichment, normalized values are used (see Methods and S1 and S2 Data). ***p<0.001, **p<0.01, *p<0.05, ns = not significant, and na = not applicable (due to detection in single proteomics experiment).

| | Protein Identified LC-MS/MS | Vertebrate Homolog | MW (kDa) | Total Experiments Enriched | Total Peptide Spectra | | | Normalized Peptide Spectra | | |
|---|---|---|---|---|---|---|---|---|---|---|
| | | | | | GS::GFP (control) | GS:: RPM-1 | GS:: RPM-1 LD | Significance GS:: RPM-1 vs GS:: GFP (p-value) | Significance GS:: RPM-1 vs GS::GFP (q-value, FDR 0.05) | RPM-1 LD vs RPM-1 Fold Increase |
| **Purification Target** | RPM-1 | MYCBP2 (PAM) | 418 | 7 | 0 | 1265 | 2874 | - | - | - |
| **Integrin Receptors** | PAT-3 | ITGB | 90 | 5 | 0 | 18 | 11 | * | 0.107 | 0.23x |
| | PAT-2 | ITGA | 136 | 2 | 0 | 7 | 0 | ns | 0.360 | 0 |
| **Kindlin Complex** | UNC-112 | FERMT1 | 82 | 5 | 9 | 33 | 29 | ** | 0.027 | 0.32x |
| | PAT-4 | ILK | 52 | 4 | 18 | 46 | 36 | * | 0.077 | 0.25x |
| | UNC-97 | LIMS1/2 | 40 | 3 | 0 | 9 | 0 | ns | 0.150 | 0 |
| | PAT-6 | PARVA | 43 | 3 | 4 | 15 | 14 | ns | 0.150 | 0.36x |
| **Talin Complex** | TLN-1 | TLN1 | 279 | 5 | 36 | 101 | 113 | ** | 0.027 | 0.47x |
| | DEB-1 | VCL | 121 | 7 | 124 | 243 | 232 | *** | 0.016 | 0.35x |
| **Other Adhesome Proteins** | PAB-1 | PABPC1 | 72 | 6 | 176 | 401 | 389 | ** | 0.020 | 0.35x |
| | PAB-2 | PABPC1 | 76 | 6 | 0 | 128 | 143 | ** | 0.020 | 0.49x |
| | FLN-1 | FLNA | 246 | 7 | 44 | 263 | 111 | ** | 0.027 | 0.1x |
| | ACT-4 | ACTB | 42 | 5 | 626 | 1408 | 879 | ** | 0.027 | 0.12x |
| | CRT-1 | CALR | 46 | 5 | 14 | 54 | 37 | ** | 0.027 | 0.19x |
| | ERM-1 | MSN (EZR) | 66 | 6 | 41 | 90 | 72 | * | 0.046 | 0.24x |
| | ATN-1 | ACTN1 | 107 | 5 | 81 | 151 | 83 | * | 0.107 | 0.07x |
| | CAV-1 | CAV1 | 26 | 3 | 0 | 10 | 0 | ns | 0.150 | 0 |
| | KIN-1 | PRKACA | 46 | 3 | 4 | 12 | 4 | ns | 0.150 | 0 |
| | RACK-1 | GNB2L1 | 36 | 3 | 115 | 245 | 190 | ns | 0.150 | 0.16x |
| | TNS-1 | TNS1 | 147 | 3 | 0 | 14 | 11 | ns | 0.150 | 0.3x |
| | CLP-1 | CAPN2 | 84 | 4 | 6 | 21 | 17 | ns | 0.193 | 0.26x |
| | RHO-1 | RHOA | 22 | 4 | 4 | 20 | 8 | ns | 0.193 | 0.13x |
| | LET-92 | PPP2CA | 36 | 5 | 0 | 14 | 9 | ns | 0.214 | 0.28x |
| | ARF-1 | ARF1 | 21 | 2 | 2 | 7 | 0 | ns | 0.360 | 0 |
| | MEC-12 | TUBA1B | 50 | 2 | 0 | 30 | 0 | ns | 0.360 | 0 |
| | NMY-1 | MYH9 | 229 | 2 | 6 | 15 | 8 | ns | 0.360 | 0.12x |
| | DYN-1 | DNM2 | 94 | 2 | 0 | 4 | 4 | ns | 0.999 | 0.19x |
| | LRP-2 | LRP1 | 540 | 2 | 0 | 0 | 17 | ns | 0.999 | 1.92x |
| | ABL-1 | ABL1 | 138 | 1 | 3 | 0 | 5 | na | na | na |
| | CED-10 | RAC1 | 21 | 1 | 0 | 3 | 0 | na | na | na |
| | COR-1 | CORO1B | 67 | 1 | 0 | 0 | 3 | na | na | na |
| | PKC-2 | PRKCA | 82 | 1 | 0 | 2 | 0 | na | na | na |
| | Y105E8A.25 | ARHGEF2 | 179 | 1 | 0 | 3 | 0 | na | na | na |

AP-proteomics using *C. elegans* identified physical associations between RPM-1 and numerous adhesome components. While the specific adhesome components that directly bind RPM-1 remain unclear, we anticipate this would only occur for a few components with indirect physical associations occurring between RPM-1 and the majority of adhesome players detected.

## Proteomic network analysis reveals adhesome components associated with RPM-1 are genetically linked to human neurobehavioral deficits

Next, we sought to integrate our RPM-1 proteomics data with bioinformatic network analysis of the adhesome. To do so, we generated a computationally predicted network of adhesome components identified in RPM-1 proteomics (Fig 2B, S2 Table). We then layered statistical significance values from RPM-1 AP-proteomic results onto the subset of adhesome components associated with RPM-1 (Fig 2B, S2 Table). Our analysis comprehensively illustrates the numerous adhesome components associated with RPM-1, how significant a hit they were in AP-proteomics, and the predicted protein-protein interaction network among these players. Our results highlight the PAT-3/UNC-112/TLN-1 adhesome axis (Fig 2A). This includes the integrin receptor cluster (β-integrin PAT-3/ITGB1, α-integrin PAT-2/ITGA), the Kindlin complex cluster (UNC-112/FERMT1, PAT-4/ILK, UNC-97/LIMS, PAT-6/PARVA), and the Talin complex cluster (TLN-1/TLN1, DEB-1/VCL) (Fig 2A and 2B). We note that *C. elegans* PAT-2 has three very similar mammalian orthologs in the adhesome: ITGA5, ITGAV and ITGA8 (S1 Table). We also identified many adhesome components associated with RPM-1 that do not have predicted protein-protein interactions in *C. elegans*. These included the Calpain protease CLP-1/CAPN1, Filamin FLN-1/FLNA, the Ezrin Moesin ortholog ERM-1/MSN, as well as several others (Fig 2B).

A recent clinical genetic study showed that mutations in *MYCBP2*, the human ortholog of *rpm-1*, cause a neurodevelopmental spectrum disorder called MDCD [33]. Patients with MDCD feature variable presentation of neurobehavioral abnormalities including developmental delay (DD), intellectual disability (ID), autism spectrum disorder (ASD) and epilepsy. Therefore, we investigated if any of the adhesome components present in our RPM-1 AP-proteomics dataset are also associated with human neurobehavioral abnormalities. By surveying the published literature, we found that 75% (24/32) of adhesome components identified in RPM-1 proteomics had human genetic variants linked to neurobehavioral abnormalities (Fig 2B; S3 Table). This included several components in the PAT-3/UNC-112/TLN-1 adhesome axis (Fig 2; S3 Table). These results highlight that RPM-1 is physically associated with an enriched group of adhesome components with genetic links to human neurodevelopmental deficits. Our findings are particularly intriguing, given the causal genetic link between MYCBP2/RPM-1 and a neurodevelopmental disorder.

## CRISPR engineering demonstrates adhesome components are expressed in mechanosensory neurons and co-localized with RPM-1 at axon termination sites

Because RPM-1 is associated with the PAT-3/UNC-112/TLN-1 adhesome axis, we wanted to determine if this axis is expressed in neurons where RPM-1 regulates axon development. RPM-1 is known to regulate axon termination by functioning cell-autonomously in the mechanosensory neurons of *C. elegans* [25,41,42]. Therefore, we investigated if PAT-3/β-integrin, UNC-112/Kindlin and TLN-1/Talin are expressed in mechanosensory neurons. To do so, we used CRISPR engineering to introduce a GFP tag at the C-terminus of PAT-3 and UNC-112 (S5 Fig). For TLN-1, we evaluated a previously generated CRISPR strain where

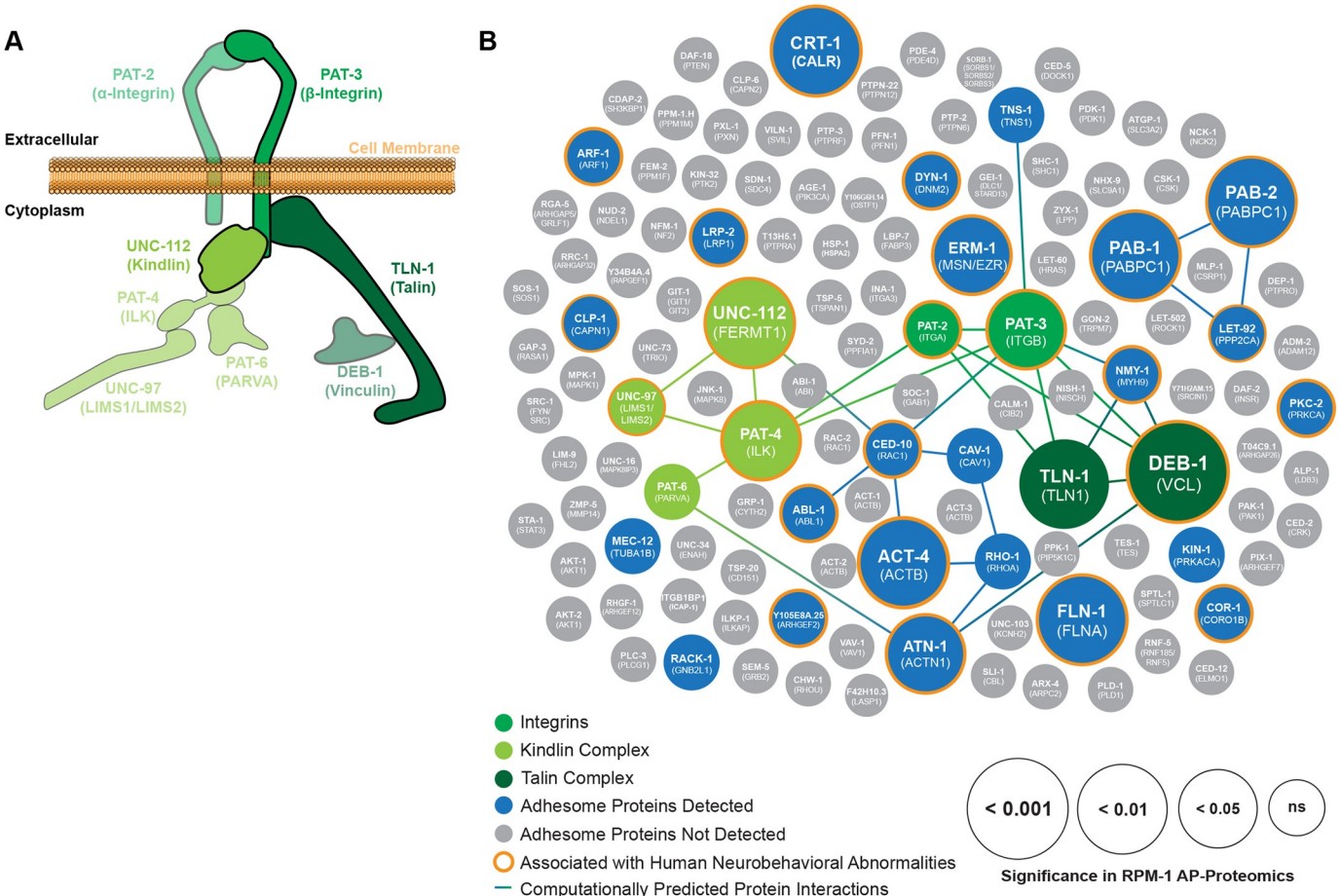

**Fig 2. PAT-3/UNC-112/TLN-1 adhesome axis and other adhesome components present in RPM-1 proteomics are linked to human neurobehavioral deficits.** **A)** Illustration of PAT-3/UNC-112/TLN-1 adhesome axis that includes two integrin receptors PAT-3 (β-Integrin, green) and PAT-2 (α-Integrin, green), the UNC-112 Kindlin complex (light green), and the TLN-1 Talin complex (dark green). **B)** Summary of RPM-1 AP-proteomics data from *C. elegans* integrated with a computationally predicted protein-protein interaction network (lines) and genetic links to human neurobehavioral abnormalities (orange halo). Adhesome components enriched in RPM-1 AP-proteomics are highlighted in color with increasing circle size denoting significance (GS::RPM-1 test samples versus GS::GFP negative controls). Highlighted (shades of green) are Integrins, UNC-112 Kindlin complex and TLN-1 Talin complex. Also shown are additional adhesome components detected (blue) or absent (gray) in RPM-1 AP-proteomics. Orthologous human protein annotated in brackets. Data is presented from 7 independent RPM-1 AP-proteomics experiments. Significance determined using Mann-Whitney test. ns = not significant.

TLN-1 was N-terminally tagged with GFP (S5 Fig) [43]. Prior studies demonstrated that null mutations in *pat-3* and *unc-112* are lethal due to the abnormal muscle formation [44]. It remains unclear if a genetic null of *tln-1* is viable because *tln-1* alleles described do not eliminate all TLN-1 isoforms (S5 Fig). Therefore, viability of animals with CRISPR GFP tags on PAT-3 and UNC-112 suggests that GFP does not impair gene function. We also used Alpha-Fold to predict the structure of CRISPR engineered PAT-3::GFP (Fig 3A), UNC-112::GFP (Fig 3B), and GFP::TLN-1 (Fig 3C). In all cases, the GFP tag was not predicted to interfere with protein folding (Fig 3A–3C). Together, structural predictions and viability of GFP knock-in strains indicate that CRISPR engineered GFP constructs are suitable for evaluating expression and localization of the PAT-3/UNC-112/TLN-1 adhesome axis.

*C. elegans* has two anterior ALM and two posterior PLM mechanosensory neurons located on the right and left side of its body (Fig 3D). Individual AVM and PVM mechanosensory

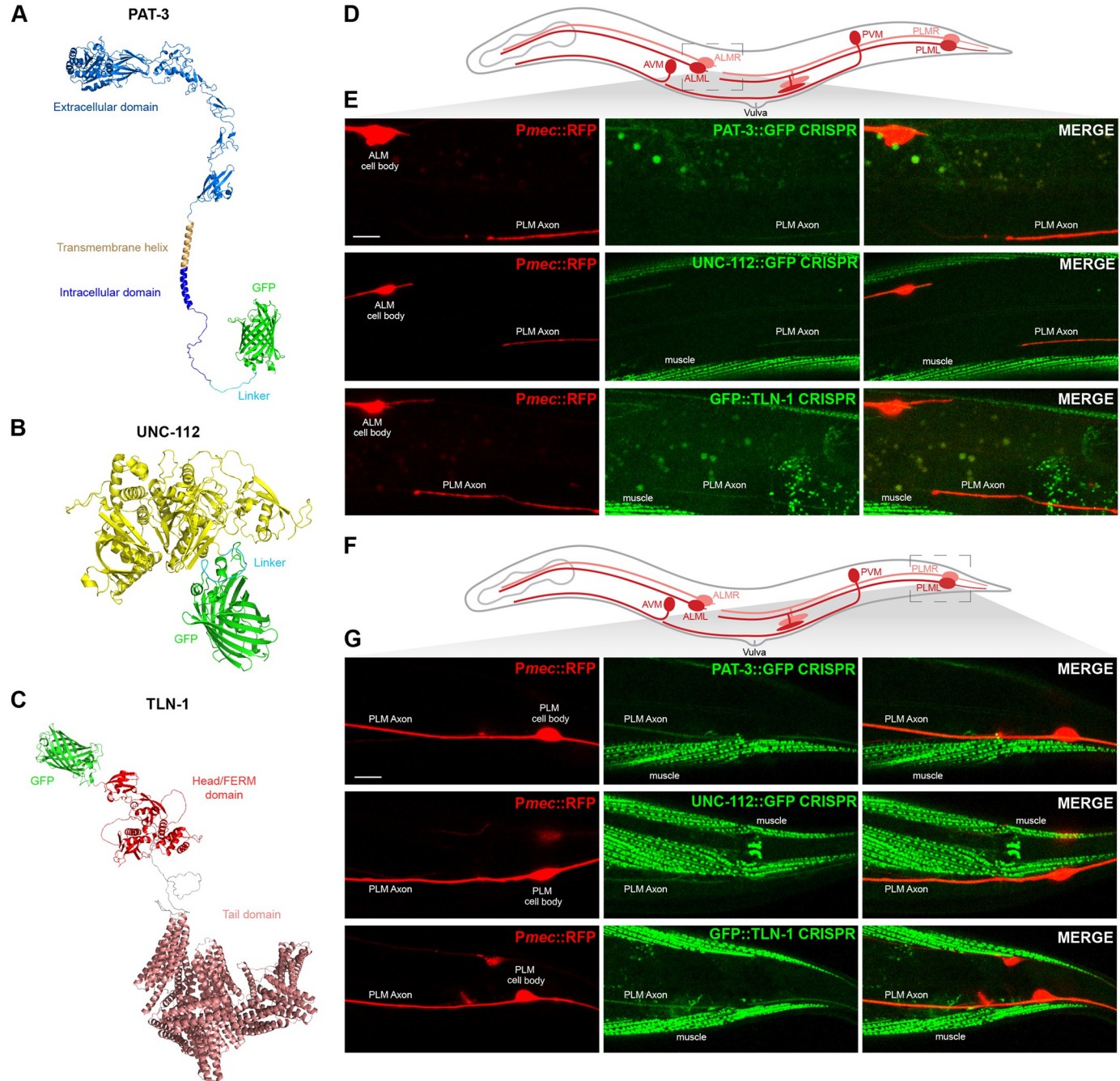

**Fig 3. PAT-3/UNC-112/TLN-1 adhesome axis is expressed in *C. elegans* mechanosensory neurons and localized to axons. A-C)** Alpha-fold predictions showing adhesome signaling components that were GFP-tagged using CRISPR engineering in *C. elegans*. Structural predictions indicate GFP is not likely to interfere with protein folding. **A)** *C. elegans* PAT-3::GFP, **B)** UNC-112::GFP, and **C)** GFP::TLN-1. **D)** Schematic of *C. elegans* mechanosensory neurons highlighting region imaged to visualize axon termination sites for PLM neurons (light gray box). **E)** Representative images showing that PAT-3::GFP, UNC-112:: GFP and GFP::TLN-1 are expressed in PLM neurons and localized to axons. P*mec-7*::mRFP (*jsIs973*) labels mechanosensory neurons. **F)** Schematic highlights cell body and initial axon segment (light gray box) of PLM neurons. **G)** Representative images showing PAT-3::GFP, UNC-112::GFP and GFP::TLN-1 primarily localized to PLM axon. For E and G, note that integrin components also show prominent expression in muscles. Scale bars 10μm.

neurons are on one side of the animal. Each ALM neuron has a soma in the midbody and sends a single axon toward the nose (Fig 3D). Each PLM soma is in the tail and has a single axon that terminates just prior to the ALM cell body (Fig 3D). We observed PAT-3::GFP, UNC-112::GFP and GFP::TLN-1 CRISPR primarily in the axons of ALM and PLM neurons (Fig 3D–3G). Further evaluation of PLM mechanosensory neurons showed that PAT-3::GFP, UNC-112::GFP and GFP::TLN-1 are localized along the length of the axon and detected at lower levels in PLM soma (Fig 3D–3G and S6 Fig). Consistent with prior studies, we also observed PAT-3::GFP, UNC-112::GFP and GFP::TLN-1 in muscles (Fig 3E and 3G). These findings with endogenous CRISPR engineered proteins show that PAT-3, UNC-112 and TLN-1 are expressed in mechanosensory neurons.

Prior studies with transgenic expression of RPM-1 in mechanosensory neurons showed that RPM-1 is concentrated at the tip of PLM axons [26, 35, 41]. To assess whether TLN-1 and RPM-1 colocalize, we used CRISPR engineering to generate animals with RPM-1::mScarlet and GFP::TLN-1. Mechanosensory neurons were labelled with transgenic mTagBFP2 (Pmec:: BFP). Super-resolution microscopy was used to image RPM-1::mScarlet and GFP::TLN-1 CRISPR. Consistent with prior findings using transgenic RPM-1, we found that endogenous RPM-1::mScarlet is concentrated at the PLM axon tip (Fig 4). Furthermore, we detected co-localization of endogenous RPM-1::mScarlet with endogenous GFP::TLN-1 at this subcellular location (Figs 4 and S7). While GFP::TLN-1 was present in the PLM axon, we also detected expression in a neurite adjacent to the PLM axon, which is most likely from the BDU neuron that forms gap junctions with the PLM axon tip [45].

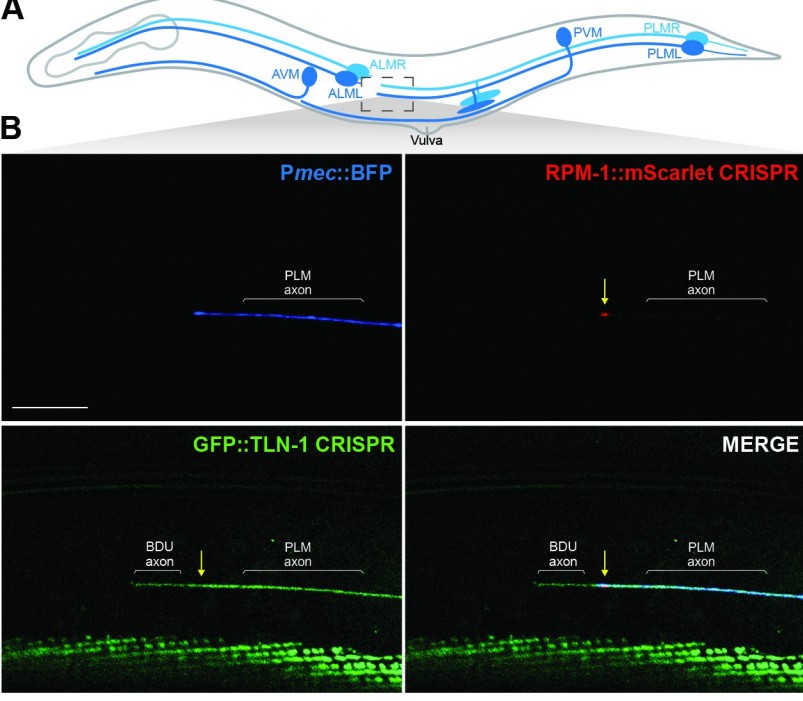

**Fig 4. TLN-1 and RPM-1 colocalize at axon termination sites in PLM mechanosensory neurons. A)** Schematic of *C. elegans* mechanosensory neurons highlighting region imaged to visualize axon termination sites for PLM neurons (light gray box). **B)** Representative super-resolution images demonstrating CRISPR engineered GFP::TLN-1 and RPM-1::mScarlet colocalize at terminated axon tips of PLM neurons. PLM axons were visualized using transgenic BFP expressed in mechanosensory neurons (Pmec17::mTagBFP, *bggEx180*). Note GFP::TLN-1 expression anterior to the PLM termination site is a neurite from the adjacent BDU neuron. Scale bar 10μm.

These CRISPR-based imaging results indicate that the endogenous PAT-3/UNC-112/TLN-1 adhesome axis is expressed in mechanosensory neurons. Furthermore, our observation that TLN-1 co-localizes with RPM-1 at PLM axon tips orthogonally validates results from proteomics indicating TLN-1 is physically associated with RPM-1.

## CRISPR-based cell-specific degradation of adhesome components results in premature axon termination in mechanosensory neurons

Having shown that the PAT-3/UNC-112/TLN-1 adhesome axis is expressed in mechanosensory neurons, we wanted to determine if genetic perturbation of this axis influences axon development. However, null mutants for *pat-3* and *unc-112* are lethal and lead to embryonic arrest at the 2-fold stage [44]. As a result, prior genetic studies in *C. elegans* have often relied upon hypomorphic mutants. To circumvent these issues, we turned to a CRISPR-based, cell-specific approach to impair individual adhesome components (Fig 5A). We combined an anti-GFP protein degradation module expressed specifically in mechanosensory neurons (*mecDEG*) [46] with PAT-3::GFP, UNC-112::GFP and GFP::TLN-1 CRISPR strains. This *mecDEG* system transgenically expresses the SOCS-box adaptor protein ZIF-1 fused with an anti-GFP nanobody. Anti-GFP nanobodies recognize CRISPR GFP-tagged adhesome components and target them for proteasome-mediated degradation by the endogenous Rbx1/Cul2 E3 ubiquitin ligase complex. This approach allowed us to circumvent lethality and evaluate how individual members of the PAT-3/UNC-112/TLN-1 adhesome axis affect axon development in mechanosensory neurons.

We chose to examine how *mecDEG*s targeting adhesome components affect PLM mechanosensory neurons because these neurons express PAT-3, UNC-112 and TLN-1 (Fig 3), and PLM neurons display precise axon termination (Fig 5B and 5C). Each PLM neuron has its soma in the tail of the animal and extends a single axon, which terminates growth posterior to the ALM cell body (Fig 5B and 5C). We began by evaluating several controls. First, we demonstrated that CRISPR tagging PAT-3, UNC-112 and TLN-1 with GFP (PAT-3::GFP, UNC-112::GFP and GFP::TLN-1) does not affect axon termination (Fig 5D; S3 Data). Second, we showed that expression of the *mecDEG* system alone does not affect axon termination (Fig 5D).

Next, we examined whether *mecDEG* co-expression with PAT-3::GFP, UNC-112::GFP or GFP::TLN-1 CRISPR affected axon termination. We observed premature axon termination defects, in which PLM axons terminated growth at or posterior to the vulva, when any of the three adhesome components are impaired (Fig 5C). Quantitation indicated that premature termination defects were significant with cell-specific degradation of each adhesome component (Fig 5D). Moderate frequency of defects occurred with degradation of PAT-3::GFP CRISPR, and defects occurred at lower frequency when UNC-112::GFP CRISPR was impaired. Interestingly, degradation of GFP::TLN-1 resulted in a particularly high incidence of premature termination defects (Fig 5D). We note that TLN-1 has 7 isoforms, and our CRISPR strategy only GFP-tagged the largest isoform, isoform a, and a truncated isoform, isoform b (S5 Fig). Therefore, our results indicate that one or both of these isoforms mediate TLN-1 function during axon termination. The particularly high frequency of defects with GFP::TLN-1 CRISPR degradation could occur for several reasons. 1) TLN-1 is a primary adhesome signaling component in axon termination. 2) TLN-1 could have additional signaling activities outside the adhesome that influence axon development. 3) Degradation of GFP::TLN-1 CRISPR by the *mecDEG* system could be more efficient than degradation of PAT-3::GFP or UNC-112::GFP.

To further evaluate the prominent premature axon termination phenotype observed when TLN-1 is impaired, we performed a transgenic rescue experiment. Here, we transgenically expressed TLN-1 using a pan-neuronal promoter in animals expressing both the *mecDEG*

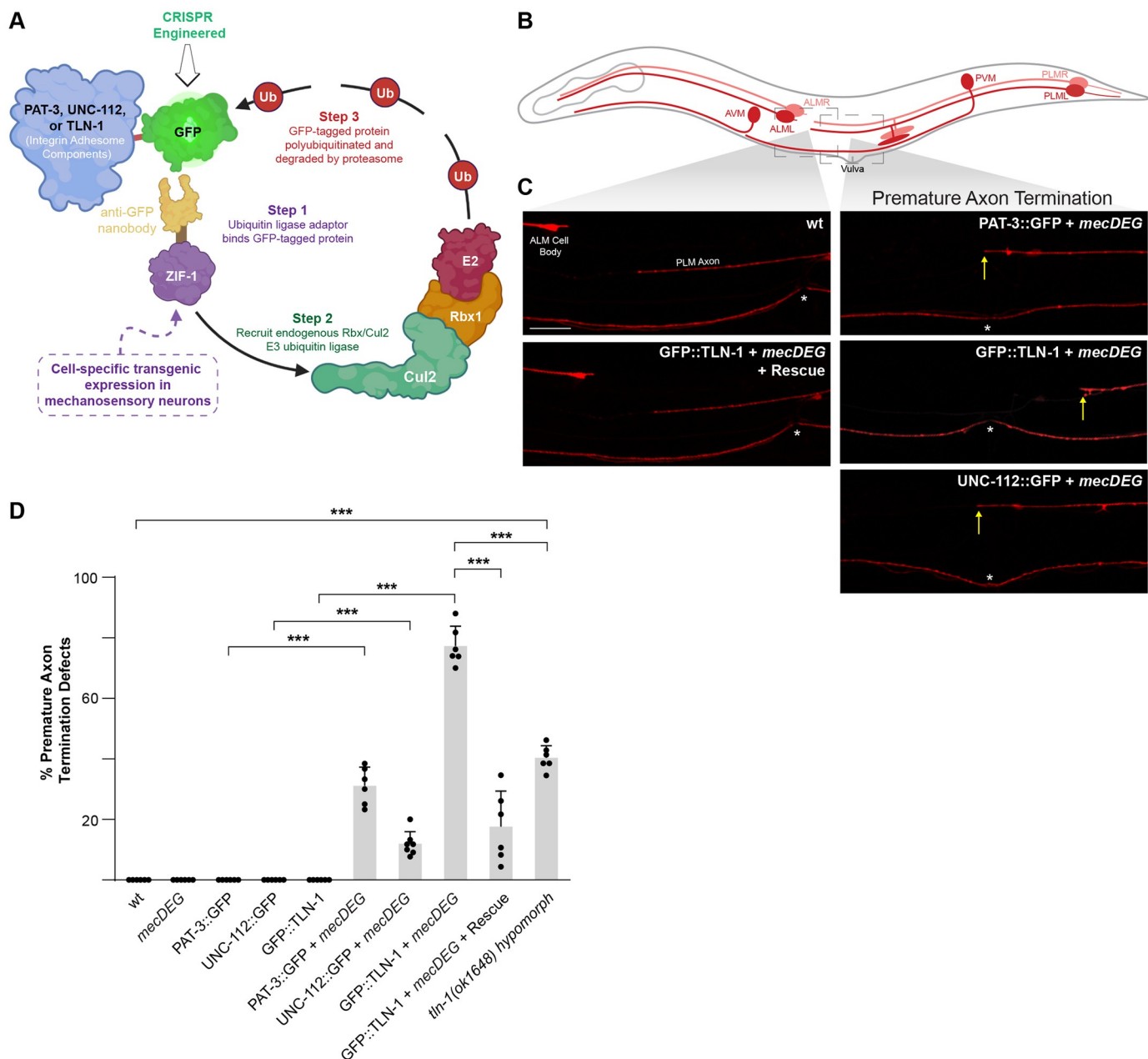

**Fig 5. Impairing PAT-3/UNC-112/TLN-1 adhesome axis with a CRISPR-based, cell-specific protein degradation system results in premature axon termination. A)** Illustration of CRISPR-based cell-specific protein degradation system used to impair adhesome components in *C. elegans* mechanosensory neurons (*mecDEG*). Adapted from Wang *et al.* 2017. **B)** Schematic of mechanosensory neurons with imaged regions highlighted (light gray boxes). **C)** Representative images of PLM axons for indicated genotypes visualized using P*mec-7*::mRFP (*jsIs973*). Premature axon termination defects (arrows) occur when *mecDEG* targets GFP that is CRISPR engineered onto adhesome components (PAT-3::GFP, UNC-112::GFP and GFP::TLN-1). Vulva (asterisks) used as anatomical reference point for premature termination. **D)** Quantitation of premature axon termination defects in PLM neurons for indicated genotypes. Note premature axon termination occurs with *mecDEG* targeting of adhesome components and in *tln-1(ok1648)* hypomorphic mutants. Means (bars) are shown for 5 or more counts (black dots) with 20 or more animals/count for each genotype. Error bars indicate SEM. Significance assessed using Student's *t*-test with Bonferroni correction for multiple comparisons. ***p<0.001. Scale bar 10μm.

system and GFP::TLN-1 CRISPR. We found that premature termination defects caused by degrading GFP::TLN-1 CRISPR were rescued by transgenic expression of untagged TLN-1, which cannot be degraded by *mecDEG* (Fig 5C). Quantitation of premature termination defects indicated that transgenic expression of TLN-1 lacking a GFP tag significantly reduced the frequency of defects (Fig 5D).

To provide further genetic validation for TLN-1 affecting axon termination, we tested a *tln-1* deletion mutant, *ok1648*. We also observed significant premature termination defects in *tln-1 (ok1648)* mutants compared to wild-type animals (Fig 5D). However, the frequency of defects was significantly lower than *mecDEG* targeting GFP::TLN-1 CRISPR. Interestingly, results from *mecDEG* targeting TLN-1 and the *tln-1* deletion mutant have helped us better understand the importance of TLN-1a and b isoforms in axon termination. *mecDEG* targeting GFP::TLN-1 CRISPR eliminates TLN-1a and b isoforms, which are the only isoforms that are N-terminally tagged with GFP (S5 Fig). In contrast, the *tln-1(ok1648)* deletion allele impairs several TLN-1 isoforms including the TLN-1a isoform, but not the TLN-1b isoform (S5 Fig). Thus, our findings suggest that TLN-1a and b isoforms are particularly important for axon termination. Moreover, our observation that *tln-1(ok1648)* mutants have lower frequency premature termination defects is likely to reflect that this allele is hypomorphic.

Collectively, our results support several conclusions. First, impairing any component in the PAT-3/UNC-112/TLN-1 adhesome axis results in premature axon termination. Second, the cell-specific *mecDEG* strategy avoids lethality associated with null alleles and demonstrates that this adhesome axis functions cell-autonomously in mechanosensory neurons to regulate axon development. Finally, our results suggest that TLN-1 could be a particularly prominent player in axon termination.

## PAT-3/UNC-112/TLN-1 adhesome axis inhibits RPM-1 to regulate axon termination

Our observations that TLN-1 colocalizes with RPM-1 at terminated axon sites and that perturbing the PAT-3/UNC-112/TLN-1 adhesome axis within mechanosensory neurons results in premature axon termination prompted us to test genetic interactions between adhesome components and RPM-1. Previous studies demonstrated that RPM-1 functions cell-autonomously in mechanosensory neurons to regulate axon termination [25,26,42]. In PLM neurons, *rpm-1* protein null mutants [25,33] result in failed axon termination where axons overgrow beyond the ALM cell body and hook towards the ventral side of the animal (Fig 6).

To assess genetic interactions between the PAT-3/UNC-112/TLN-1 axis and RPM-1, we combined *mecDEG* targeting PAT-3::GFP, UNC-112::GFP or GFP::TLN-1 CRISPR with an *rpm-1* loss of function (lf) allele. As a control, we combined *rpm-1* mutants with the *mecDEG* system, which did not alter the frequency of failed axon termination defects caused by *rpm-1* (lf) (Fig 6C; S4 Data).

Consistent with earlier experiments (Fig 5D), *mecDEG* targeting of PAT-3::GFP, UNC-112::GFP or GFP::TLN-1 CRISPR resulted in premature termination defects, the opposite defect to what occurs in *rpm-1* mutants (Fig 6B and 6C). Opposing phenotypes suggest that RPM-1 and the adhesome could be functionally opposed during axon termination.

Next, we evaluated double mutants for *rpm-1* and PAT-3/UNC-112/TLN-1 adhesome axis components (e.g. *rpm-1*; PAT-3::GFP + *mecDEG*). Double mutants principally displayed failed axon termination defects similar to *rpm-1* single mutants (Fig 6B). Quantitation indicated that premature termination defects primarily seen when adhesome components were impaired individually were significantly reduced in frequency in double mutants with *rpm-1* (Fig 6C). These results demonstrate that *rpm-1* genetically suppresses impaired PAT-3/UNC-112/TLN-

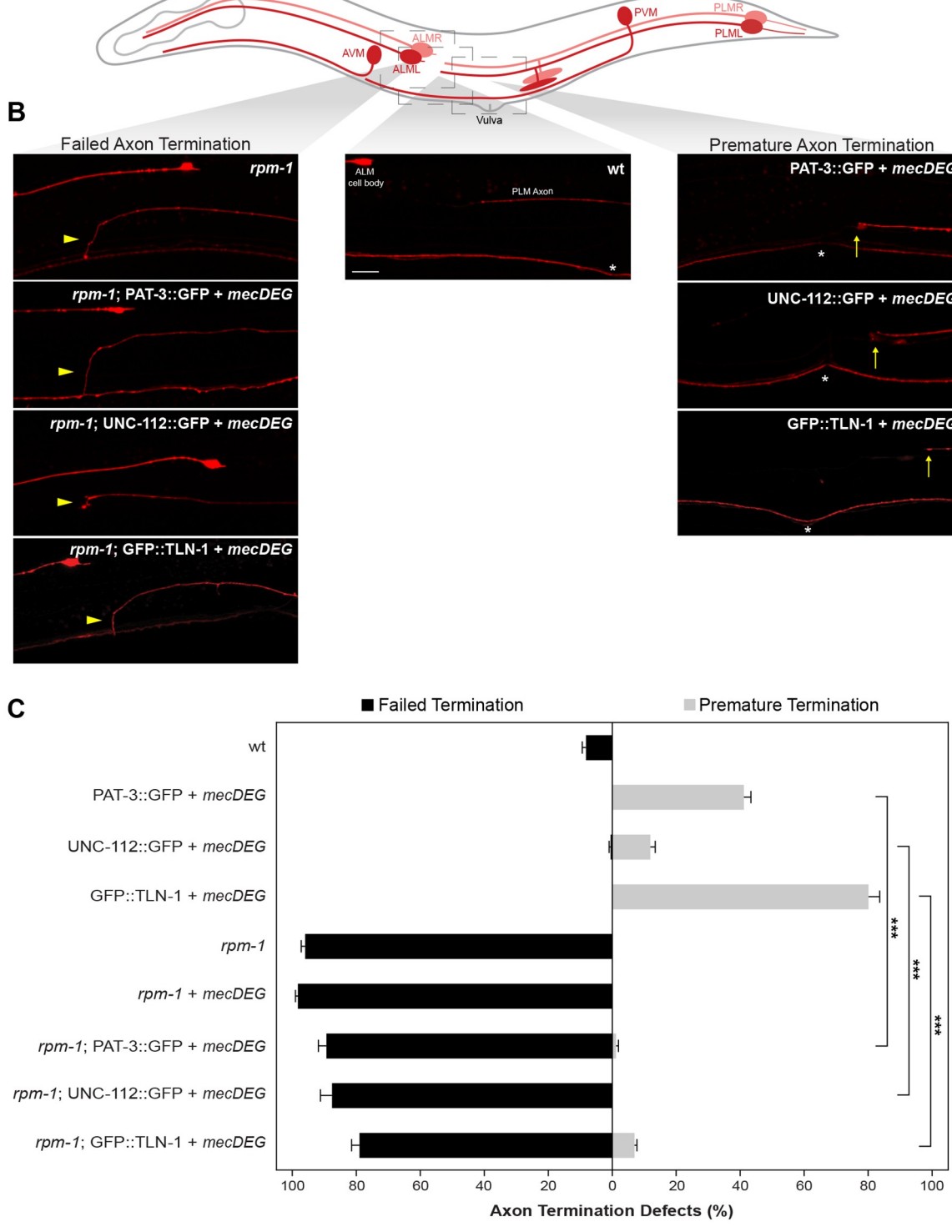

**Fig 6. PAT-3/UNC-112/TLN-1 adhesome axis inhibits RPM-1 to regulate axon termination. A)** Schematic of *C. elegans* mechanosensory neurons with imaged regions highlighted (light gray boxes). **B)** Representative images of PLM axons for indicated genotypes visualized using P*mec-7*::mRFP (*jsIs973*). *rpm-1* mutants display failed axon termination defects (arrowhead) and *mecDEG* targeting of adhesome components (PAT-3::GFP, UNC-112::GFP and GFP::TLN-1 CRISPR) results in premature termination defects (arrows). *rpm-1* double mutants with adhesome component degradation (e.g. *rpm-1;* PAT-3::GFP + *mecDEG*) predominantly display failed termination defects. Vulva (asterisks) used as anatomical reference point for premature termination. **C)** Quantitation of failed

axon termination defects (black) and premature termination defects (light gray) in PLM neurons for indicated genotypes. Premature termination defects are strongly suppressed in double mutants lacking RPM-1 and PAT-3, UNC-112 or TLN-1 adhesome components. Mean (bars) are shown for 5 or more counts (20 or more animals/count) for each genotype. Error bars indicate SEM. Significance assessed using Student's *t-test* with Bonferroni correction. ***p<0.001. Scale bar 10μm.

1 axis function. Our findings suggest that the PAT-3/UNC-112/TLN-1 adhesome axis inhibits RPM-1 to facilitate termination of axon outgrowth.

## Talin inhibits RPM-1 to regulate growth cone development during axon termination

Our prior work demonstrated that RPM-1 regulates axon termination by promoting growth cone collapse in PLM mechanosensory neurons [41]. Therefore, we performed developmental time-course experiments to examine whether genetic interactions between *rpm-1* and the PAT-3/UNC-112/TLN-1 adhesome axis affects growth cones during termination of axon outgrowth. For developmental studies, we focused on Talin because *mecDEG* targeting of TLN-1 yielded the highest incidence of premature termination defects (Figs 5 and 6), suggesting it could be a key adhesome component involved in axon termination.

Prior developmental time-course studies with *rpm-1* loss-of-function (lf) mutants proved extremely challenging for several reasons [41]. These experiments require precise synchronization of animal development, extensive imaging sessions, and evaluation of extremely small larval stage animals (Fig 7A, L1-L3). As a result, it was not technically feasible to perform developmental studies with all the control strains tested in axon termination studies on adults (Figs 5D and 6C). Therefore, we evaluated a more limited but essential set of controls. This included GFP::TLN-1 CRISPR alone as a negative control, because we observed no differences in axon termination for this control, *mecDEG* alone or wild-type animals (Fig 5D). We opted to evaluate *rpm-1* (lf) mutants carrying the *mecDEG* control as we observed no differences in axon termination defects between this strain and *rpm-1* mutants lacking *mecDEG* (Fig 6).

We began by evaluating growth cone frequency and size at different developmental time points across all larval stages—L1, L2, L3 and L4 (Fig 7A). Several observations were consistent with prior studies. Growth cone frequency decreased between 1h and 16h post-hatch (PH) in control GFP::TLN-1 CRISPR control animals (Fig 7B–7E; S5–S7 Data). This occurred with similar timelines and decreasing growth cone frequency as previously observed for wild-type animals [41]. Decreasing growth cone frequency was accompanied by termination of axon outgrowth which occurred between 7 and 16h PH (Fig 7B and 7G). Based on our scoring criteria, failed termination defects occurred in GFP::TLN-1 controls at moderate frequency at 16h PH. However, this reflects a normal part of development, as overextension is alleviated over time when animals increase in size and form gaps between ALM cell bodies and PLM axon termination sites [41,47]. Thus, overextension present at this time point is not due to CRISPR engineering GFP onto TLN-1.

In *rpm-1* single mutants carrying *mecDEG* alone (*rpm-1 + mecDEG*), growth cone collapse was impaired resulting in both increased growth cone frequency (Fig 7B–7E) and increased growth cone size (Fig 7B and 7F; S8 Data) compared to GFP::TLN-1 controls. Due to impaired growth cone collapse, failed axon termination defects emerge at high frequency in *rpm-1 + mecDEG* by 16h PH (L2) and persist through young adult animals (44h PH) (Fig 7B and 7G; S9 Data). Our findings with *rpm-1 + mecDEG* are consistent with prior findings with *rpm-1* (lf) mutants [41]. Our developmental findings are also consistent with the incidence of failed termination defects being similar in *rpm-1* (lf) mutants and *rpm-1 + mecDEG* mutants

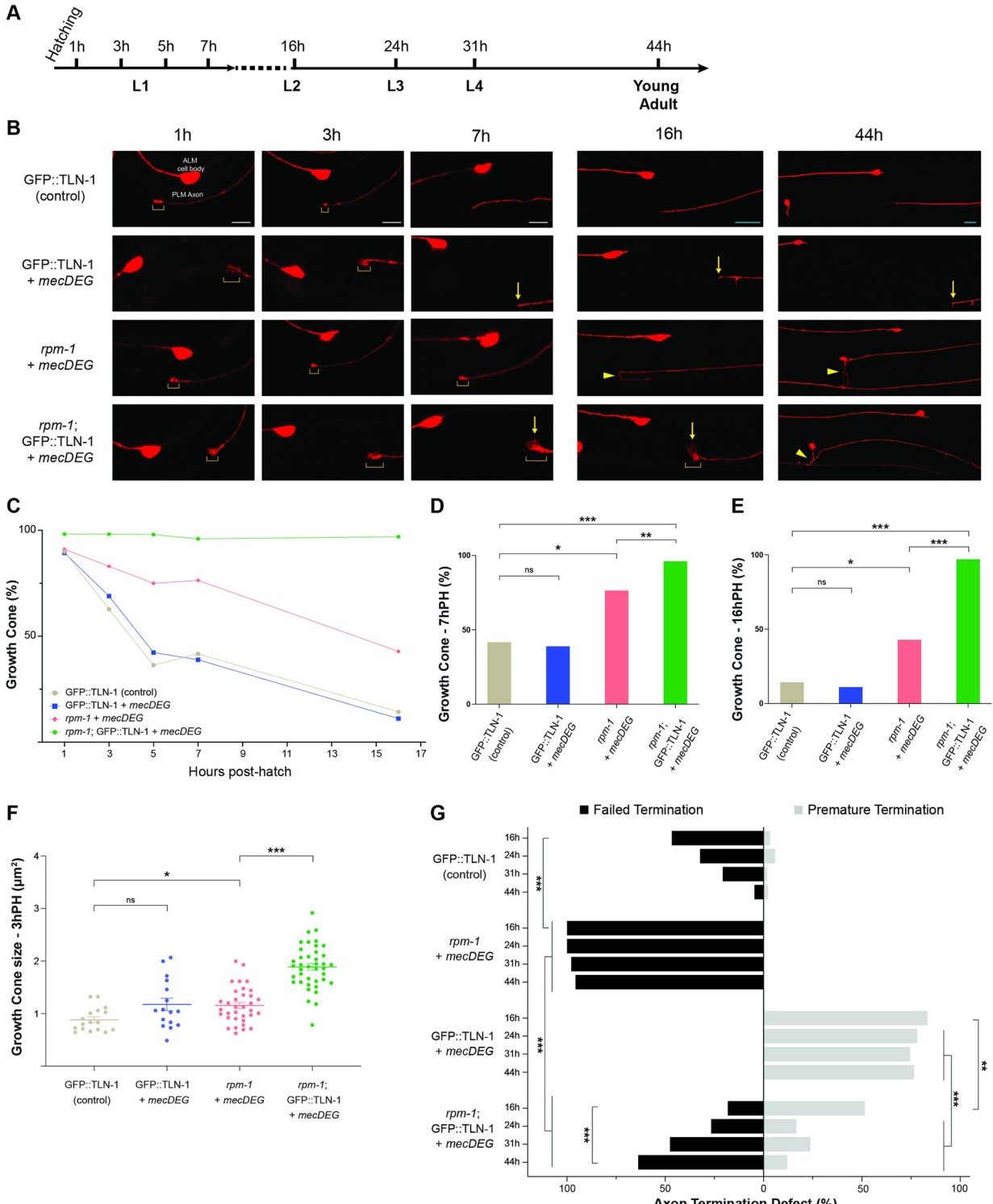

**Fig 7. Time-course studies indicate TLN-1 inhibits RPM-1 to regulate growth cone collapse and axon termination during development. A)** Timeline of *C. elegans* larval development and key time points evaluated. **B)** Representative images of PLM axons for indicated genotypes visualized using P*mec-7*::mRFP (*jsIs973*). Shown are axonal growth cones (brackets), premature termination sites (arrows) and failed termination defects (arrowheads). Double mutants (*rpm-1;* GFP::TLN-1 + *mecDEG*) display failed growth cone collapse with persistent, enlarged growth cones. In double mutants, growth cones are initially in locations corresponding to premature axon termination (1 to 16h PH) but failed axon termination becomes the primary phenotype as development progresses (44h PH). Note AVM cell body is only visible at 44h PH on one side of

the animal. **C)** Summary of quantitative results for growth cone frequency during development for indicated genotypes. **D-E)** Quantitative analysis of growth cone frequency at **D)** 7h PH and **E)** 16h PH. **F)** Quantitation of growth cone size at 3h PH for indicated genotypes. **G)** Quantitation of failed axon termination defects (black) and premature termination defects (light gray) in PLM neurons for indicated genotypes at specified time points in development. For C-G, minimum of 21 PLM neurons scored for each time point and genotype. Mean (square, bar or line) are shown for each genotype. For F, dots represent a single animal. Error bars indicate SEM. For C-E and G, significance assessed using Fisher's exact test. For F, significance assessed using Student's *t-test* with Bonferroni correction for growth cone size. ns = not significant, *p<0.05, **p<0.01, ***p<0.001. Scale bars 5μm (1-7h PH, white bars) or 10μm (16-44h PH, teal bars).

(Fig 6C). Taken together, these results indicate that the *mecDEG* system does not further influence growth cone development beyond the effects of *rpm-1* (lf) in the absence of an endogenous CRISPR-tagged GFP target protein.

We then evaluated GFP::TLN-1 + *mecDEG* animals (*tln-1* single mutants), and examined whether premature growth cone collapse leads to premature axon termination defects. GFP::TLN-1 + *mecDEG* did not have altered growth cone frequency (Fig 7B–7E) or growth cone size (Fig 7B and 7F) when compared to GFP::TLN-1 CRISPR control animals. GFP::TLN-1 + *mecDEG* animals displayed growth cones in locations corresponding with premature termination as early as 1h PH (Fig 7B), and premature termination defects accumulated between 7 and 16h PH (Fig 7B and 7G).

Next, we evaluated *rpm-1*; GFP::TLN-1 + *mecDEG* animals (*rpm-1* and *tln-1* double mutants). We observed increased growth cone frequency (Fig 7B–7E) and size (Fig 7B and 7F) in *rpm-1*; GFP::TLN-1 + *mecDEG* double mutants compared to *rpm-1* + *mecDEG* single mutants. Growth cones persisted at high frequency out to 16h PH (Fig 7B, 7C and 7E). Interestingly, growth cones that persist at 16h PH in *rpm-1*; GFP::TLN-1 + *mecDEG* double mutants present as a mixture of potential failed and premature termination events based on anatomical locations where growth cones are present at this time point (Fig 6B and 6G). While axon termination has not occurred at 16hr PH in double mutants, we applied the same scoring scheme used to score axon termination in adults. As development progresses, potential premature termination in *rpm-1*; GFP::TLN-1 + *mecDEG* double mutants is overcome and the failed termination phenotype present in *rpm-1* single mutants primarily takes over between 24 and 44h PH (Fig 6B and 6G).

Our results indicate that in *rpm-1*; GFP::TLN-1 + *mecDEG* double mutants the main phenotypes across axon development are impaired growth cone collapse, increased growth cone size, and ultimately failed axon termination. These are all primary phenotypes in *rpm-1* single mutants. Thus, our results indicate that multiple *rpm-1* (lf) phenotypes are present while premature termination caused by *tln-1* (lf) is suppressed in double mutants. Collectively, these findings further support the model that TLN-1 functions as an upstream inhibitor of RPM-1.

## TLN-1 and RPM-1 influence axon termination via effects on microtubule stability

We previously used genetic and pharmacological approaches to show that the RPM-1 ubiquitin ligase signaling hub regulates growth cone collapse and axon termination, in part, by acting as a microtubule destabilizer [41]. Our genetic results here indicate the PAT-3/UNC-112/TLN-1 adhesome axis inhibits RPM-1. If this is the case, we would expect that altered axon termination caused by impairing TLN-1 might be sensitive to colchicine, a microtubule destabilizing drug.

To test this, we began by treating wild-type animals with 0.25mM and 0.5mM colchicine. While 0.25mM colchicine did not affect termination, 0.5 mM colchicine resulted in significant premature termination defects compared to sham (DMSO) treated animals (Fig 8A–8C;

S10 Data). Consistent with prior findings [41], both 0.25mM and 0.5mM colchicine significantly reduced the frequency of failed termination defects in *rpm-1* (lf) mutants (Fig 8B and 8C). Thus, increased microtubule stability in *rpm-1* (lf) mutants leads to axon overgrowth, which can be rescued by the colchicine microtubule destabilizer.

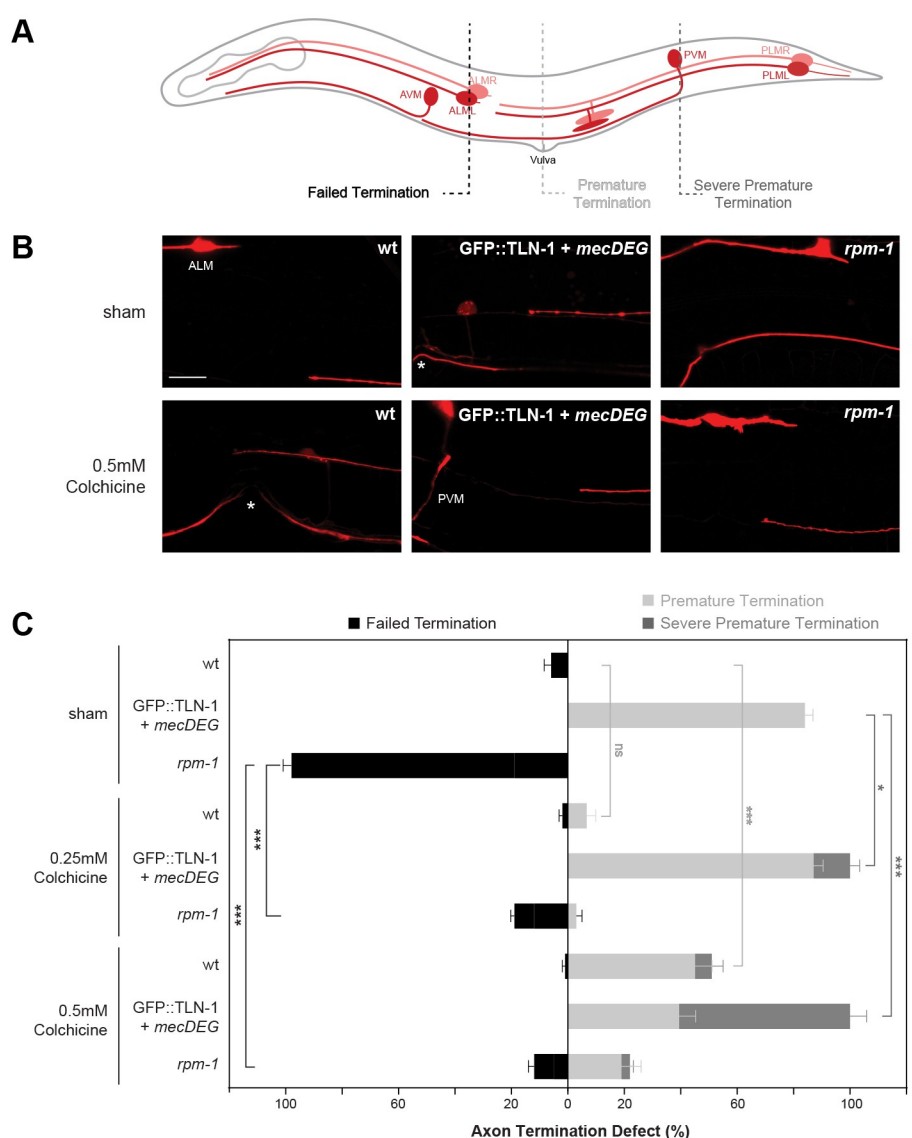

**Fig 8. TLN-1 and RPM-1 influence axon termination via effects on microtubule stability. A)** Schematic of *C. elegans* mechanosensory neurons. Dashed lines indicate different regions where axon termination defects were visualized. **B)** Representative images of PLM axons for indicated genotypes under sham conditions (DMSO) or treated with the microtubule destabilizing drug colchicine (0.5mM). Mechanosensory neurons visualized using P*mec-7*::mRFP (*jsIs973*). **C)** Quantitation of failed termination (black), premature termination (light gray) and severe premature termination (dark gray) defects for indicated genotypes and treatment. *rpm-1* mutants display failed axon termination defects that are suppressed by colchicine. GFP::TLN-1 degraded by *mecDEG* results in premature termination defects with colchicine treatment resulting in more severe premature termination defects. Vulva (asterisks) and PVM are anatomical reference points for premature and severe premature termination defects, respectively. Mean (bars) are shown for 5 counts (20 or more animals/count) for each genotype and treatment. Error bars indicate SEM. Significance assessed using Student's *t-test* with Bonferroni correction. ns = not significant, *p<0.05, **p<0.01, ***p<0.001. Scale bar 10μm.

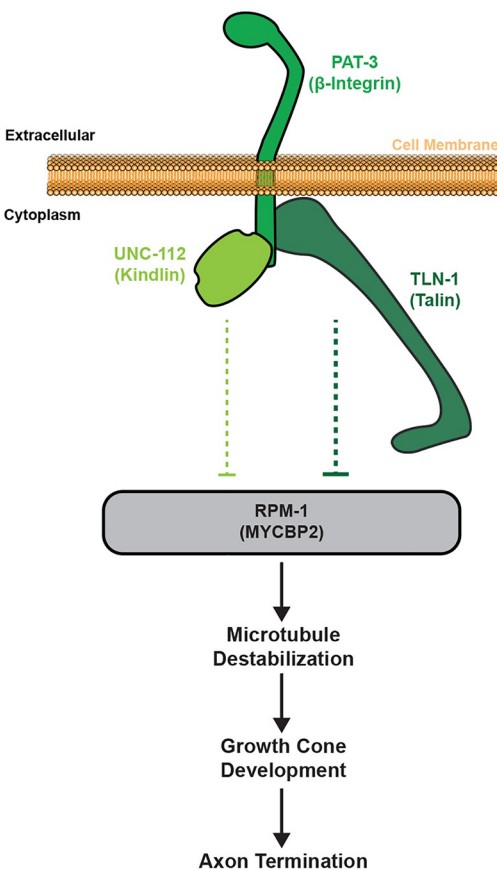

**Fig 9. Summary of results indicating PAT-3/UNC-112/TLN-1 adhesome axis inhibits RPM-1 to influence microtubules and growth cone dynamics during axon termination.**

In contrast, we observed an increase in the severity of premature termination defects when GFP::TLN-1 + *mecDEG* animals were treated with colchicine (Fig 8B). Premature termination was considered severe if the PLM axon terminated outgrowth at or posterior to the PVM cell body. Quantitation indicated that both 0.25mM and 0.5mM colchicine significantly increased the frequency of severe premature termination defects in GFP::TLN-1 + *mecDEG* animals (Fig 8C). A corresponding decrease in premature termination defects that occur at the vulva also occurred. Thus, coordinately applying colchicine and impairing TLN-1 increases microtubule destabilization leading to more severe premature termination.

Our findings support two conclusions. First, RPM-1 and TLN-1 both affect axon termination by influencing microtubule stability. Second, our results indicate RPM-1 and TLN-1 having opposing effects on microtubule stability during termination. These observations are both consistent with TLN-1 inhibiting RPM-1 to influence microtubule stability and axon termination. Importantly, these outcomes with pharmacological manipulation of microtubule stability further support our model that the PAT-3/UNC-112/TLN-1 adhesome axis inhibits RPM-1 to regulate axon termination (Fig 9).

## Discussion

Bioinformatic and proteomic studies have identified an enormous integrin signaling network called the adhesome, which is evolutionarily conserved from humans through *C. elegans*

[8,39]. Here, we have independently updated and verified the *C. elegans* adhesome. *C. elegans* provides us with an organismal setting to evaluate the composition and function of a simplified adhesome, and allows us to test the functional relationship between the adhesome and prominent players in nervous system development. Three core components of the putative *C. elegans* adhesome are PAT-3/β-integrin, UNC-112/Kindlin and TLN-1/Talin. Prior studies have principally examined their roles in body wall muscle development, which leads to lethality in null mutants [44,48]. Much less is known about how these adhesome components influence nervous system development in *C. elegans*. A small number of studies have used hypomorphic alleles to show that integrins regulate axon guidance in motor neurons [12,49,50]. At present, we know particularly little about how Kindlin and Talin shape axon development *in vivo*.

Here, we have taken a first step towards obtaining a more comprehensive, mechanistic understanding of how the adhesome shapes axonal and nervous system development. To do so, we integrated proteomics with cell-specific genetic tools to evaluate how the PAT-3/UNC-112/TLN-1 adhesome axis shapes axon development in the mechanosensory neurons of *C. elegans*. CRISPR engineering and targeted degradation of PAT-3, UNC-112 and TLN-1, circumvented lethality caused by global null alleles. Our results demonstrate that this adhesome axis is expressed (Fig 3) and functions cell-autonomously within mechanosensory neurons to regulate termination of axon outgrowth (Fig 5).

A collective series of observations indicate that the PAT-3/UNC-112/TLN-1 adhesome axis regulates axon termination by inhibiting the RPM-1 ubiquitin ligase signaling hub. **1)** Large-scale *C. elegans* proteomics with RPM-1 identified numerous adhesome components including the PAT-3/UNC-112/TLN-1 adhesome axis physically associated with RPM-1 (Fig 1, Table 1). **2)** *In vivo* super-resolution imaging in *C. elegans* where CRISPR engineering was used to label molecular components demonstrated that GFP::TLN-1 co-localizes with RPM-1::mScarlet at axon termination sites (Fig 4). **3)** Genetic interaction studies indicate that premature termination defects caused by impairing components in the PAT-3/UNC-112/TLN-1 adhesome axis are suppressed by *rpm-1* (lf) (Figs 5 and 6). **4)** Developmental time-course studies indicate that when RPM-1 and TLN-1 are simultaneously impaired we observe failed growth cone collapse, increased growth cone size and failed axon termination, which are the primary phenotypes caused by *rpm-1* (lf) (Fig 7). **5)** Finally, outcomes from pharmacological studies indicate that RPM-1 and TLN-1 have opposing effects on microtublue dynamics leading to opposing outcomes on axon termination. This provides a further mechanistic explanation for how PAT-3/UNC-1112/TLN-1 inihibiton of RPM-1 influences microtubules to affect axon termination. We note the caveat that the cell-specific CRISPR-based degradation approach we use to target components in the PAT-3/UNC-112/TLN-1 adhesome axis does not generate null alleles. Nonetheless, the most reasonable interpretation of our collective findings is that the PAT-3/UNC-112/TLN-1 axis inhibts RPM-1 to affect microtubule dynamics, growth cone collapse and termination of axon outgrowth (Fig 9). Importantly, our findings also now provide orthogonal, unbiased identification and validation of the adhesome using proteomic, genetic and developmental studies in a whole organism setting.

Developmental studies were valuable in evaluating the cellular mechanism underpinning the inhibitory genetic relationship between the PAT-3/UNC-112/TLN-1 adhesome axis and RPM-1. Our results indicate the PAT-3/UNC-112/TLN-1 axis promotes initial axon outgrowth, as axons terminate prematurely when these components are impaired compared to CRISPR engineered GFP controls or wild-type animals (Figs 5–7). Our findings with PAT-3 in *C. elegans* mechanosensory neurons are consistent with prior studies indicating β-integrins can regulate axon outgrowth in cultured neurons and axons of the corpus callosum in rodents [16,51]. Our observation that UNC-112 is required for axon development in a whole animal setting expands upon prior studies that used shRNA to knockdown Kindlin in cultured

neurons [20,52]. The third adhesome component we tested, TLN-1, has not been previously shown to shape axon development in any system. Our observation that premature termination defects occur with higher frequency when TLN-1 is perturbed compared to UNC-112 suggests that Talin could be a more prominent adhesome component in the axon termination process (Fig 5). This supports recent commentary (based on studies outside the nervous system) that Talin could be a master player in adhesome signaling [17]. However, we note our results indicate Talin and Kindlin function are both needed for accurate, efficient axon termination.

To provide further mechanistic insight into how PAT-3/UNC-112/TLN-1 axis inhibition of RPM-1 affects axon termination, we turned to impacts on the microtubule cytoskeleton. Our results indicate that TLN-1 and RPM-1 have opposing functional relationships with microtubules. RPM-1 is known to destabilize microtubules with excess microtubule stability in *rpm-1* mutants contributing to failed axon termination and overgrowth (Fig 8) [41]. We found that the severity of premature termination defects is increased when TLN-1 is perturbed, and microtubules are further destabilized by colchicine indicating TLN-1 functions to increase microtubule stability (Fig 8). This finding is consistent with TLN-1 inhibiting RPM-1. Indeed, our results would suggest that TLN-1 inhibition of RPM-1 and its destabilizing effects on microtubules might be a mechanism by which TLN-1 affects microtubule stability in axons. While β-integrin, Kindlin and Talin interact with and influence actin, cellular studies have provided less widely recognized but important evidence that links these adhesome components to microtubule stability. For example, cell-based results indicate that integrin/FAK/Rho signaling stabilizes microtubules at the leading edge of migrating cells [53]. Likewise, cell-based studies and biochemical findings indicate Talin interacts with KANK1 to affect microtubule stabilization [54]. Finally, Kindlin has been shown to regulate microtubule acetylation which stabilizes microtubules [55]. Our observations suggest that the PAT-3/UNC-112/TLN-1 adhesome axis inhibits RPM-1 to influence microtubule stability *in vivo* (Fig 9).

Our findings here have also now revealed further links between RPM-1 signaling and the adhesome. One example is the Cdk5 kinase which phosphorylates Talin and is inhibited by RPM-1 ubiquitin ligase activity [35,56]. The MIG-15/NIK kinase interacts genetically with RPM-1 and physically binds PAT-3 β-Integrin [49,57]. Integrins and Kindlin affect axon regeneration and axon degeneration, functional contexts that also involve RPM-1 and its murine ortholog Phr1 [4,28,58–61]. Finally, studies in mice indicate that loss of Phr1 or β-Integrin result in developmental defects in the corpus callosum [51,62,63]. Our results now provide genetic, developmental and molecular mechanisms that potentially explain these previously unrecognized links between the PAT-3/UNC-112/TLN-1 adhesome axis and RPM-1 signaling.

There is growing evidence that integrin signaling is involved in brain disorders [2,3]. By combining proteomics and computational network analysis, we identified human genetic variants associated with neurobehavioral abnormalities (*i.e.* intellectual disability, developmental delay or autism) in 75% of adhesome components physically associated with RPM-1 (Fig 2). This is a particularly interesting observation, as a recent study identified multiple *de novo* human variants in *MYCBP2/rpm-1* that were shown to be deleterious by CRISPR editing in *C. elegans* [33]. Patients with *MYCBP2* variants have a neurodevelopmental disorder called MDCD that features corpus callosum defects and a spectrum of neurobehavioral deficits including developmental delay, intellectual disability and autism. Thus, we have identified an adhesome subnetwork that is associated with neurobehavioral abnormalities, and that interacts physically and functionally with a gene that causes a neurodevelopmental disorder. Our findings now point to adhesome biology as a potentially rich direction for future studies on nervous system development and disease.

## Methods

### Genetics and strains

*C. elegans* N2 isolate was used for all experiments. Animals were maintained using standard procedures. The following mutant alleles were used: *rpm-1*(*ju44*) V and *tln-1*(*ok1648*) I. The following integrated transgene was used: *jsIs973* [$P_{mec-7}$::mRFP] III. The following MosSCI transgene was used: *itSi953* [$P_{mec-18}$::vhhGFP4::ZIF-1::operon-linker::mKate::tbb-2 3'UTR + Cbr-unc-119(+)] II, which we refer to as *mecDEG*. The MosSCI transgene was inserted into *ttTi5605* on LG II. The following CRISPR alleles were used: *unc-112*(*bgg68* [UNC-112::GFP CRISPR]), *pat-3*(*bgg86* [PAT-3::GFP CRISPR]), *tln-1*(*zh117* [GFP::TLN-1 CRISPR]), *rpm-1* (*bgg119* [RPM-1::mScarlet CRISPR]). The following extrachromosomal arrays were used: *bggEx172* ($P_{rgef-1}$::FLAG::TLN-1) and *bggEx180* ($P_{mec-17}$::mTagBFP2). We sequenced the *tln-1* (*ok1648*) allele and verified that is affects TLN-1 isoform a, c, d and e but does not affect the coding sequence for TLN-1 isoform b and f. This independently confirms what is annotated on WormBase (https://wormbase.org).

All transgenes and CRISPR alleles used for specific experiments are described in S4 Table. All mutants and transgenic lines were outcrossed four or more times prior to experiments. Animals were grown at 23°C for genetic analysis. Sequences of all genotyping primers for mutants and CRISPR strains can be found in S5 Table.

### Molecular biology and Transgenics

Five fragments of the *tln-1* genomic DNA (gDNA) (pBG-454 to pBG-458) were amplified from N2 genomic DNA using iProof High-Fidelity DNA polymerase (BioRad) or Q5 High-Fidelity DNA Polymerase (NEB). Fragments were then assembled into a single *tln-1* gDNA clone in a pCR8 vector (pBG-GY1088) using HiFi DNA assembly (NEB). Using gateway technology (LR clonase II, Invitrogen), *tln-1* gDNA was recombined with a plasmid containing $P_{rgef-1}$::FLAG and *unc-54* 3'UTR (PBG-GY134) to generate the final $P_{rgef-1}$::FLAG::*tln-1* plasmid (pBG-GY1100) that was co-microinjected with $P_{rps-27}$::NeoR (pBG-264) to generate the TLN-1 rescue *bggEx172* ($P_{rgef-1}$::FLAG::TLN-1, $P_{rps-27}$::NeoR). mTagBFP2 was PCR amplified from the OH15495 strain using Q5 High-Fidelity DNA Polymerase (NEB), followed by TOPO cloning into PCR8 (pBG-GY1120). We then used multisite gateway LR recombination with pDEST R4-R3, pDONR P4-P1R (pBG-GY823), pCR8 + mTagBFP2 (pBG-GY1120), and pDONR P2R-P3 with let-858 3'UTR (pBG-GY827) to generate the final $P_{mec-17}$::mTagBFP2 plasmid (pBG-GY1121) that was co-microinjected with $P_{ttx-3}$::RFP (pBG-41) to generate *bggEx180* ($P_{mec-17}$::mTagBFP2, $P_{ttx-3}$::RFP).

All constructs were fully sequenced, and transgenic extrachromosomal arrays were generated using standard microinjection procedures for *C. elegans*. Injection conditions and transgene construction details are specified in S6 Table.

### CRISPR/Cas9 engineering

CRISPR alleles were engineered utilizing *dpy-10* co-CRISPR or *rol-6* plasmid co-injection, and direct injection of *Cas9* ribonucleoprotein complexes. Injection mixes contained tracrRNA (IDT), repair template (PCR or ssODN (Ultramers DNA oligo IDT)), and recombinant Cas9 protein purified from Rosetta 2 *E. coli* (Millipore EMD, #71397). Ribonucleoprotein complexes were placed at 37°C for 15 minutes prior to injection. For RPM-1:: mScarlet, a *C. elegans* codon-optimized mScarlet-I with 3 artificial introns was PCR amplified from pMS050 (Addgene Plasmid #91826). For PAT-3::GFP CRISPR and RPM-1::mScarlet CRISPR, a hybrid repair template was utilized [64]. Gene edits were confirmed by

sequencing, and edited strains were outcrossed four or more times to N2 animals. All CRISPR injection conditions are shown in S6 Table. All crRNA and repair template sequences are shown in S7 Table.

## C. elegans proteomics

Detailed methodology for *C. elegans* AP-proteomics was previously described [26,34]. In brief, experiments were conducted on mixed-stage animals transgenically expressing GS::RPM-1, GS::RPM-1 LD substrate 'trap' or GS::GFP (negative control). Animals were harvested, separated from culture debris by sucrose flotation centrifugation, frozen in liquid nitrogen, and ground into submicron particles using a cryomill (Retsch). Samples were then lysed under various detergent conditions. Resulting whole-animal lysates were centrifuged to separate particulate material. Lysates were incubated with IgG-coupled Dynabeads (80 mg total protein with 500uL of beads) at 4°C for 4 hours. Samples were run on Tris-Glycine SDS-PAGE gels, followed by Coomassie staining and in-gel trypsin digested. The resulting peptide pools were acidified, desalted, dried, and subsequently resuspended in 100 μl of 0.1% formic acid. Samples were run on an Orbitrap Fusion Tribrid Mass Spectrometer (ThermoFisher Scientific) coupled to an EASY-nLC 1000 system and on-line eluted on an analytical RP column (0.075 × 250 mm Acclaim PepMap RLSC nano Viper, ThermoFisher Scientific). Scaffold (Proteome Software) was used to analyze MS/MS data to identify peptides and proteins in samples. All control and test samples for single proteomics experiments were blinded for genotype and underwent mass spectrometry together.

We performed a total of 7 independent AP-proteomics experiments with RPM-1 (see S1 and S2 Data). Identification of adhesome components physically associated with RPM-1 was based on three criteria: 1) Detection in one or more proteomic experiments. 2) 1.5x or greater total spectra enrichment in GS::RPM-1 or GS::RPM-1 LD sample compared to GS::GFP negative control. 3) Elimination of ribosomal and vitellogenin proteins, as they are widely recognized as proteomic contaminants.

Mass spectrometry files for RPM-1 AP-proteomics (Project number: PXD051783) have been uploaded to the PRIDE database (https://www.ebi.ac.uk/pride/).

## PAT-3, UNC-112, TLN-1 expression and co-localization with RPM-1 in mechanosensory neurons

Young adult PAT-3::GFP, UNC-112::GFP, or GFP::TLN-1 CRISPR animals with the transgene *jsIs973* [P$_{mec-7}$::mRFP] were anesthetized using 10 μM levamisole. Animals were mounted on 3% agarose pads on glass slides with coverslips. Z-stack images were captured using a Zeiss LSM 710 microscope equipped with 40x oil-immersion objective.

For colocalization studies, we examined young adult GFP::TLN-1 CRISPR; RPM-1::mScarlet CRISPR animals carrying transgenic extrachromosomal arrays containing P$_{mec-17}$:: mTagBFP2 that labels mechanosensory neurons. Animals were anesthetized using 10 μM levamisole, and mounted on 3% agarose pads on glass slides with coverslips. Super-resolution imaging was performed on single image slices using a Zeiss LSM 900 microscope with Airyscan 2 super-resolution under a 63x oil-immersion objective.

## Analysis of axon termination

We evaluated axon termination in PLM neurons using the transgene *jsIs973* [P$_{mec-7}$::mRFP]. We considered four termination zones based on anatomical criteria when scoring animals. 1) Wild-type axon termination was considered to occur anterior to the vulva but at or posterior to the ALM soma. 2) Failed axon termination defects occurred when axons overgrew and

overextended anterior to the ALM soma and/or formed ventral 'hooks' that occurred anterior to the vulva or anterior to the ALM cell body. 3) Premature termination defects occurred if the PLM axon terminated growth at or posterior to the vulva. 4) Severe premature termination defects occurred if the PLM axon terminated growth posterior to the PVM axon.

To quantify axon termination, young adult animals were anesthetized using 10 μM levamisole in M9 buffer. Animals were mounted onto 2% agar pads on glass slides with coverslips. Animals were visualized using a Leica DM5000 B (CTR5000) epifluorescent microscope equipped with a 40x oil-immersion objective.

### Developmental analysis

The transgene *jsIs973* (P*mec-7*::mRFP) was used to visualize growth cone frequency and growth cone size in PLM neurons and to measure the distance between PLM axon termination sites and ALM somas.

Developmental time-course studies were performed as previously described [41]. Animals were synchronized by collecting freshly hatched L1 larvae every 10–15 minutes and were then cultured on plates to desired time points at 23˚C. At select time points between 1 and 44 h PH, animals were mounted in 5μM levamisole (1-16h PH) or 10μM levamisole (24-44h PH) in M9 buffer on 3% agarose pads. Images were acquired under 63× (1-31h PH) or 40× (44h PH) magnification on a Zeiss LSM 710 laser scanning confocal microscope.

Image analysis of growth cone width and area as well as growth cone distance from the ALM soma was executed using Fiji/ImageJ software from NIH image. We defined growth cones compared to terminated axon tips using both morphological assessment and quantitative measurements of the ratio of growth cone width to axon width as previously described [41]. A growth cone was defined as having a ratio of growth cone width to axon width of 1.5x or greater. Images of all genotypes and time points were blinded and mixed before analysis using custom software run with Python. We used criteria above to categorize termination defects in adults (44h PH) when a vulva could be observed and used as an anatomical reference point. For L1-L4 larval stages (16h to 31h PH) where a vulval was not present, we defined axon termination defects (failed or premature) by measuring the distance between the middle of the ALM soma to the end of the PLM axon (growth cone or terminated axon tip). Premature termination was classified depending on the distance between ALM and terminated axon tip or growth cone during larval stages as follows: 15μm for 1, 3, 5, 7h PH, 24μm for 16h PH, 55μm for 24h PH, 65μm for 31h PH. These metrics were derived based on control animals for a given time point. Failed termination was defined as an axon terminating anterior to the ALM soma or the presentation of a 'hook' towards the ventral cord for all stages.

### Microtubule pharmacology

Pharmacological manipulation of microtubule stability was performed as previously described [41]. We prepared 3.5cm NGM plates with colchicine (0.25mM and 0.5mM), or DMSO (sham control). These drugs were applied to the plates and allowed to diffuse overnight. The following day, plates were seeded with OP50 *E. coli* and bacterial lawns grown overnight. 3 P$_0$ adult animals were transferred onto drug-treated or control plates, allowed to lay eggs overnight and were removed. Colchicine resulted in reduced brood size at 0.25mM and 0.5mM. F$_1$ progeny that remained healthy after drug treatment were scored for axon termination as young adults.

### AlphaFold prediction

Structures of GFP fusion proteins were predicted using the LocalColabFold version of ColabFold 1.5.0 using pdb templates with "—templates" and relaxed using "—amber" [65]. Five

models were predicted for each protein and the model with the highest predicted local distance difference test (plDDT) was presented for each protein. Predicted structures were adjusted and colorized in PyMOL (PyMOL Molecular Graphics System, Version 2.5.0, Schrödinger, LLC). The initial output for GFP::TLN-1 resembled one of its templates: the closed, autoinhibited conformation of human TALIN 1 (PDB 6R9T). For visual clarity, overlapping domains were manually separated by adjusting dihedral angles in residues with pLDDT scores equal to 50 or less and lacking secondary structure. Thus, the presented model represents a partially open conformation where the head and tail regions show their predicted compact forms but do not interact. PAT-3::GFP was also similarly adjusted to separate the intracellular and extracellular domains.

## *C. elegans* adhesome computational network analysis

Based on a list of 232 molecular components that comprise the mammalian adhesome network [8,9], we sought to identify all potential *C. elegans* orthologs based on four primary criteria: 1) 30% or higher sequence similarity; 2) 20% or higher sequence identity; 3) prior annotation in the *C. elegans* adhesome [39], which we carefully re-evaluated here ourselves; and 4) examination of reverse BLAST top hit annotations for human orthologs determined using the Alliance of Genome Resources database (https://www.alliancegenome.org) and secondary verification using WormBase (https://wormbase.org). We determined that 132 mammalian adhesome components have *C. elegans* orthologs, and of these 123 mammalian components have unique, single orthologous components in *C. elegans*.

Utilizing STRING software [66], these 123 conserved, unique *C. elegans* adhesome components were mapped into a computational network of predicted protein-protein interactions. Line connections between proteins indicate prior evidence for protein-protein interactions. Predicted protein-protein interactions were based on experiments and/or databases, and we reported outcomes with a minimum interaction score of 0.4. Gene ontology analysis was also conducted using STRING.

RPM-1 AP-proteomics data was overlaid onto the predicted *C. elegans* adhesome signaling network. Any orthologous adhesome protein detected in RPM-1 AP-proteomics is shown in blue (Fig 2B). Their overall enrichment in GS::RPM-1 test samples over GS::GFP negative control samples are represented by the size of the circle with increasing size indicating greater significance.

**Statistical analysis.** *Proteomics*: For AP-proteomics, protein hits that were identified with 1.5x or greater enrichment in GS::RPM-1 or GS::RPM-1 LD test samples over GS::GFP negative control samples were subjected to unpaired, nonparametric Mann-Whitney tests for significance using normalized total spectral counts across seven independent proteomic experiments (see S1 and S2 Data). These comparisons are reported as p values. To correct for multiple comparisons, we applied a 5% false discovery rate (FDR) and a *post hoc* Benjamini-Hochberg method to Mann-Whitney tests. These comparisons are reported as q values. Each experiment was considered an individual n for analysis. For statistical comparisons in Table 1, we normalized total spectral counts for the candidate hit protein to both its molecular weight as well as the total spectral counts and molecular weight of the affinity purification target, as previously detailed [34]. Similar normalization for protein size was done for calculation of fold enrichment of candidates between GS::RPM-1 and GS::RPM-1 LD samples in Table 1.

*Axon termination*: For analysis of PLM axon termination, statistical comparisons were performed using unpaired, two-tailed Student's *t*-test with Bonferroni correction for multiple comparisons. Statistical analysis was performed using GraphPad Prism software. Error bars

are SEM. Significance was defined as p < 0.05. Bar graphs represent averages from 4 to 10 counts (25–35 neurons/count) for each genotype from three or more independent experiments. Dots in plots represent averages for single-counts.

*Developmental analysis*: For growth cone frequency in PLM neurons, statistical comparisons were made by Fisher's exact test using GraphPad Prism software. Significance was defined as p < 0.05. Bar graphs represent averages from at least 21 animals obtained from 2 or more, independent imaging session for each genotype.

For growth cone size in PLM neurons, statistical comparisons were made by Student's *t*-test with Bonferroni correction for multiple comparisons using GraphPad Prism software. Error bars are SEM. Significance was defined as p < 0.05. Lines in scatter plots represent averages from at least 16 animals obtained from 2 or more, independent imaging session for each genotype. Dots shown in plots represent values from each individual animal.

For axon termination in PLM neurons, statistical comparisons were made by Fisher's exact test using GraphPad Prism software. Significance was defined as p < 0.05. Bar graphs represent averages from at least 23 animals obtained from 2 or more, independent imaging session for each genotype.

## Supporting information

**S1 Fig. Representation of adhesome components in top 20 gene ontology process terms for RPM-1 proteomic hits generated using STRING.**
(TIF)

**S2 Fig. PAT-3 peptide coverage in RPM-1 proteomics, related to Fig 1.** Highlighted are 8 peptides in PAT-3 (red) identified in GS::RPM-1 and GS::RPM-1 LD samples.
(TIF)

**S3 Fig. UNC-112 peptide coverage in RPM-1 proteomics, related to Fig 1.** Highlighted are 17 peptides in UNC-112 (red) identified in GS::RPM-1 and GS::RPM-1 LD samples.
(TIF)

**S4 Fig. TLN-1 peptide coverage in RPM-1 proteomics, related to Fig 1.** Highlighted are 51 peptides in TLN-1 (red) identified in GS::RPM-1 and GS::RPM-1 LD samples.
(TIF)

**S5 Fig. CRISPR engineering GFP tags onto PAT-3, UNC-112 and TLN-1, related to Fig 3.** Protein diagrams with annotated protein domains highlight location of CRISPR engineered GFP tag on PAT-3, UNC-112 and TLN-1. Note that only two TLN-1 isoforms, TLN-1 a and b, are tagged with GFP. Also annotated is *tln-1* hypomorphic deletion allele *ok1648* (red box), which only affects TLN-1 isoforms a, c, d, and e.
(TIF)

**S6 Fig. Adhesome components are localized at low levels to PLM soma, related to Fig 3. A)** Schematic of PLM neurons with imaged regions highlighted (light gray box). **B)** Representative images showing that PAT-3::GFP, UNC-112::GFP and GFP::TLN-1 are localized to PLM soma at low levels compared to axon localization. PAT-3::GFP and GFP::TLN-1 images acquired at 40x magnification and UNC-112::GFP images acquired at 100x magnification. Scale bar 10μm.
(TIF)

**S7 Fig. Further examples of TLN-1 and RPM-1 colocalization at PLM axon termination sites, related to Fig 4.** Shown are multiple examples of super-resolution images demonstrating

CRISPR engineered GFP::TLN-1 and RPM-1::mScarlet colocalize at terminated axon tips of PLM neurons. PLM axons were visualized using transgenic BFP expressed in mechanosensory neurons (P*mec-17*::BFP, *bggEx180*). Note GFP::TLN-1 expression anterior to the PLM termination site is neurite from adjacent BDU neuron. Scale bar 10μm.
(TIF)

**S1 Table. Annotation of human and *C. elegans* adhesome components (excel).**
(XLSX)

**S2 Table. List of STRING computationally predicted protein-protein interactions for adhesome components identified in RPM-1 proteomics, related to Fig 2 (excel).**
(XLSX)

**S3 Table. List of human genetic variants in adhesome components associated with neurobehavioral abnormalities, related to Fig 2 (excel).**
(XLSX)

**S4 Table C. *elegans* strain list, related to methods (word).**
(DOCX)

**S5 Table. Genotyping and cloning primers, related to methods (word).**
(DOCX)

**S6 Table. Injection conditions, related to methods (word).**
(DOCX)

**S7 Table. crRNA and repair template sequences, related to methods (word).**
(DOCX)

**S1 Data. Analysis of mass spectrometry data for adhesome components identified in 7 independent RPM-1 AP-proteomics experiments, related to Figs 1 and 2 (excel).**
(XLSX)

**S2 Data. Statistical analysis of mass spectrometry data for adhesome components identified in 7 independent RPM-1 AP-proteomics experiments, related to Figs 1 and 2 (prism).**
(PZFX)

**S3 Data. Data set related to Fig 5D (prism).**
(PZFX)

**S4 Data. Data set related to Fig 6C (prism).**
(PZFX)

**S5 Data. Data set related to Fig 7C (prism).**
(PZFX)

**S6 Data. Data set related to Fig 7D (prism).**
(PZFX)

**S7 Data. Data set related to Fig 7E (prism).**
(PZFX)

**S8 Data. Data set related to Fig 7F (prism).**
(PZFX)

**S9 Data. Data set related to Fig 7G (prism).**
(PZFX)

**S10 Data. Data set related to Fig 8C (prism).**
(PZFX)

## Acknowledgments

We would like to thank Dr. George Tsaprailis from the Proteomics Core at the University of Florida Scripps Labs (formerly The Scripps Research Institute—Florida) for his aid in analysis of mass spectra. We thank the *C. elegans* Genetics Center for strains and the WormBase genetic resource database.

## Author Contributions

**Conceptualization:** Jonathan Amezquita, Muriel Desbois, Melissa A. Borgen, Brock Grill.

**Data curation:** Jonathan Amezquita, Muriel Desbois, Karla J. Opperman, Joseph S. Pak, Elyse L. Christensen, Karen Diaz-Garcia.

**Formal analysis:** Jonathan Amezquita, Muriel Desbois, Elyse L. Christensen.

**Funding acquisition:** Brock Grill.

**Investigation:** Jonathan Amezquita, Muriel Desbois, Karla J. Opperman, Elyse L. Christensen, Karen Diaz-Garcia.

**Methodology:** Jonathan Amezquita, Muriel Desbois, Karla J. Opperman, Joseph S. Pak, Nikki T. Nguyen, Melissa A. Borgen.

**Project administration:** Brock Grill.

**Supervision:** Brock Grill.

**Writing – original draft:** Jonathan Amezquita, Muriel Desbois, Brock Grill.

**Writing – review & editing:** Jonathan Amezquita, Muriel Desbois, Karla J. Opperman, Joseph S. Pak, Melissa A. Borgen, Brock Grill.

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
