## [Editor Report · Decision Letter 0]

14 May 2024

Dear Dr Grill,

Thank you very much for submitting your Research Article entitled 'Axon development is regulated at genetic and proteomic interfaces between the integrin adhesome and the RPM-1 ubiquitin ligase signaling hub' to PLOS Genetics.

The manuscript was previously reviewed by peer reviewers for PLOS Biology and has now been fully evaluated at the editorial level by PLOS Genetics. The reviewers appreciated the attention to an important problem, but raised some substantial concerns about the current manuscript. Based on the existing reviews, we will not be able to accept this version of the manuscript, but we would be willing to review a much-revised version. We cannot, of course, promise publication at that time.

The editors who have evaluated this manuscript particularly advise firming up the molecular epistasis as suggested by reviewer #3 during peer review at PLOS Biology. While additional biochemical experiments are not essential for further consideration at PLOS Genetics, it would be great to see at least one interaction confirmed independently.

Should you decide to revise the manuscript for further consideration here, your revisions should address the specific points made by each reviewer from PLOS Biology. We will also require a detailed list of your responses to the review comments and a description of the changes you have made in the manuscript.

If you decide to revise the manuscript for further consideration at PLOS Genetics, please aim to resubmit within the next 60 days, unless it will take extra time to address the concerns of the reviewers, in which case we would appreciate an expected resubmission date by email to plosgenetics@plos.org.

Please do not hesitate to contact us if you have any concerns or questions.

Yours sincerely,

Bing Ye

Guest Editor

PLOS Genetics

Aimée Dudley

Editor-in-Chief

PLOS Genetics

---

## [Editor Report · Decision Letter 1]

12 Nov 2024

Dear Dr Grill,

We are pleased to inform you that your manuscript entitled "Integrin adhesome axis inhibits the RPM-1 ubiquitin ligase signaling hub to regulate growth cone and axon development" has been editorially accepted for publication in PLOS Genetics. Congratulations!

Yours sincerely,

Bing Ye

Guest Editor

PLOS Genetics

Aimée Dudley

Editor-in-Chief

PLOS Genetics

Comments from the reviewers (if applicable):

**Data Deposition**

http://datadryad.org/submit?journalID=pgenetics&manu=PGENETICS-D-24-00392R1

**Press Queries**

---

## [Editor Report · Acceptance letter]

25 Nov 2024

PGENETICS-D-24-00392R1 

Integrin adhesome axis inhibits the RPM-1 ubiquitin ligase signaling hub to regulate growth cone and axon development 

Dear Dr Grill, 

We are pleased to inform you that your manuscript entitled "Integrin adhesome axis inhibits the RPM-1 ubiquitin ligase signaling hub to regulate growth cone and axon development" has been formally accepted for publication in PLOS Genetics! Your manuscript is now with our production department and you will be notified of the publication date in due course.

With kind regards,

Dorothy Lannert

PLOS Genetics

On behalf of:
